# *PHAROH* lncRNA regulates Myc translation in hepatocellular carcinoma via sequestering TIAR

**Allen T Yu[1,2], Carmen Berasain[3,4,5], Sonam Bhatia[1], Keith Rivera[1], Bodu Liu[1], Frank Rigo[6], Darryl J Pappin[1], David L Spector[1,2]\***

[1]Cold Spring Harbor Laboratory, Cold Spring Harbor, United States; [2]Genetics Program, Stony Brook University, Stony Brook, United States; [3]Hepatology Program, Cima, University of Navarra, Pamplona, Spain; [4]Instituto de Investigaciones Sanitarias de Navarra-IdiSNA, Pamplona, Spain; [5]CIBERehd, Instituto de Salud Carlos III, Madrid, Spain; [6]Ionis Pharmaceuticals, Carlsbad, United States

**Abstract** Hepatocellular carcinoma, the most common type of liver malignancy, is one of the most lethal forms of cancer. We identified a long non-coding RNA, *Gm19705*, that is overexpressed in hepatocellular carcinoma and mouse embryonic stem cells. We named this RNA *Pluripotency and Hepatocyte Associated RNA Overexpressed in HCC*, or *PHAROH*. Depletion of *PHAROH* impacts cell proliferation and migration, which can be rescued by ectopic expression of *PHAROH*. RNA-seq analysis of *PHAROH* knockouts revealed that a large number of genes with decreased expression contain a *Myc* motif in their promoter. MYC is decreased in knockout cells at the protein level, but not the mRNA level. RNA-antisense pulldown identified nucleolysin TIAR, a translational repressor, to bind to a 71-nt hairpin within *PHAROH*, sequestration of which increases MYC translation. In summary, our data suggest that *PHAROH* regulates MYC translation by sequestering TIAR and as such represents a potentially exciting diagnostic or therapeutic target in hepatocellular carcinoma.

**\*For correspondence:** spector@cshl.edu

## Introduction

Hepatocellular carcinoma (HCC), the most common type of liver malignancy, is one of the most lethal forms of cancer (*Asrani et al., 2019*). HCC is the fifth most frequently diagnosed cancer and the third leading cause of cancer-related deaths worldwide (*Villanueva, 2019*). The molecular landscape of HCC is very complex and includes multiple genetic and epigenetic modifications that could represent new diagnosis and therapeutic targets. In this sense, multiple studies have established molecular classifications of HCC subtypes that could be related to clinical management and outcomes (*Dhanasekaran et al., 2019*; *Llovet et al., 2018*). For instance, Hoshida et al. classified HCC into S1, S2, and S3 subtypes by means of their histological, pathological, and molecular signatures (*Hoshida et al., 2009*). S1 tumors exhibit high TGF-β and Wnt signaling activity but do not harbor mutations or genomic changes. The tumors are relatively large, poorly differentiated, and associated with poor survival. S2 tumors have increased levels of Myc and phospho-Akt and overexpress α-fetoprotein, an HCC serum biomarker. S3 tumors harbor mutations in *CTNNB1* (β-catenin) but tend to be well-differentiated and are associated with good overall survival.

The standard of care for advanced HCC is treatment with sorafenib, a multi-kinase inhibitor that targets Raf, receptor tyrosine kinases (RTKs), and the platelet-derived growth factor receptor (PDGFR). Sorafenib extends the median survival time from 7.9 months to 10.7 months, and lenvatinib, a multiple VEGFR kinase inhibitor, has been reported to extend survival to 13.6 months (*Llovet et al., 2018*; *Philip et al., 2005*; *Rimassa and Santoro, 2009*). Combination therapies of

VEGF antagonists together with sorafenib or erlontinib are currently being tested (*Dhanasekaran et al., 2019*; *Greten et al., 2019*; *Quintela-Fandino et al., 2010*). However, even with the most advanced forms of treatment, the global death toll per year reaches 700,000, creating a mortality ratio of 1.07 with a 5-year survival rate of 18% (*Ferlay et al., 2010*; *Siegel et al., 2014*; *Villanueva, 2019*). Not only is it difficult to diagnose HCC in the early stages, but there is also a poor response to the currently available treatments. Thus, novel therapeutic targets and treatments for HCC are urgently needed.

The ENCODE consortium revealed that as much as 80% of the human genome can be transcribed, while only 2% of the genome encodes for proteins (*Djebali et al., 2012*). Thousands of transcripts from 200 nucleotides (nt) to over 100 kilobases (kb) in length, called long non-coding RNAs (lncRNAs), are the largest and most diverse class of non-protein-coding transcripts. They commonly originate from intergenic regions or introns and can be transcribed in the sense or antisense direction. Most are produced by RNA polymerase II and can be capped, spliced, and poly-adenylated (reviewed in *Rinn and Chang, 2012*). Strikingly, many are expressed in a cell- or tissue-specific manner and undergo changes in expression level during cellular differentiation and in cancers (*Costa, 2005*; *Dinger et al., 2008*). These lncRNAs present as an exciting class of regulatory molecules to pursue as some are dysregulated in HCC and have the potential to be specific to a subtype of HCC (*Li et al., 2015*).

One of the few examples of a lncRNA that has been studied in the context of HCC is the homeobox (HOX) antisense intergenic RNA (*HOTAIR*). This transcript acts in trans by recruiting the Polycomb repressive complex 2 (PRC2), the lysine-specific histone demethylase (LSD1), and the CoREST/REST H3K4 demethylase complex to their target genes (*Ezponda and Licht, 2014*). *HOTAIR* promotes HCC cell migration and invasion by repressing RNA binding motif protein 38 (RBM38), which is otherwise targeted by p53 to induce cell cycle arrest in G1 (*Shu et al., 2006*; *Yu et al., 2015*). Another mechanism through which lncRNAs function involves inhibitory sequestration of miRNAs and transcription factors (*Cesana et al., 2011*). In HCC, the lncRNA *HULC* (highly upregulated in liver cancer) sequesters *miR-372*, which represses the protein kinase PRKACB, and downregulates the tumor suppressor gene *CDKN2C* (p18) (*Wang et al., 2010*). Similarly, the highly conserved *MALAT1* lncRNA controls expression of a set of genes associated with cell proliferation and migration and is upregulated in many solid carcinomas (*Amodio et al., 2018*; *Lin et al., 2007*); siRNA knockdown of *MALAT1* in HCC cell lines decreases cell proliferation, migration, and invasion (*Lai et al., 2012*).

Only a small number of the thousands of lncRNAs have been characterized in regard to HCC. Therefore, whether and how additional lncRNAs contribute to HCC remains unknown, and it is not fully understood how lncRNAs acquire specificity in their mode of action at individual gene loci. A lack of targetable molecules limits the effectiveness of treatments for HCC, and this class of regulatory RNAs has great potential to provide novel therapeutic targets.

Here, we reanalyzed naïve and differentiated transcriptomes of mouse embryonic stem cells (ESCs) in the context of the GENCODE M20 annotation. We aimed to identify lncRNAs that are required for the pluripotency gene expression program, and dysregulated in cancer, with a specific focus on HCC. Since normal development and differentiation are tightly regulated, dysfunction of potential regulatory RNAs may lead to various disease phenotypes including cancer. One lncRNA that is highly upregulated in HCC is of special interest, and we show that it interacts with and sequesters the translation repressor nucleolysin TIAR, resulting in an increase of Myc translation. Together, our findings identified a mechanism by which a lncRNA regulates translation of MYC in HCC by sequestering a translation inhibitor and as such has potential as a therapeutic target in HCC.

## Results

### Deep sequencing identifies 40 lncRNAs dysregulated in ESCs and cancer

Since normal development and differentiation are tightly regulated processes, we reasoned that lncRNAs whose expressions are ESC specific and can be found to also exhibit altered expression in cancer may have important potential roles in regulating critical cellular processes.

We reanalyzed the raw data from our published differential RNA-seq screen comparing lncRNA expression in mouse ESCs vs. neural progenitor cells (NPCs) (*Bergmann et al., 2015*), using updated bioinformatic tools and the recently released GENCODE M20 annotation (January 2019), which has nearly 2.5 times more annotated lncRNAs than the previously used GENCODE M3. Principal component analysis (PCA) of the processed data showed that ESCs and NPCs independently cluster, and the difference between ESC cell lines (AB2.2) and mouse-derived ESCs only accounted for 4% of the variance (*Figure 1A*). Additionally, we prioritized transcripts with an FPKM value >1, and those that were more than twofold upregulated in ESCs compared to NPCs. This left us with 147 ESC-specific transcripts. Since our goal is to discover novel transcripts that may play a role in the progression of human cancer, we first needed to identify the human homologues of the 147 mouse ESC transcripts. In addition to sequence conservation, we also evaluated syntenic conservation of the mouse lncRNAs to the human genome due to the fact that many lncRNAs are not conserved on the sequence level. Finally, we queried TCGA databases via cBioportal to find lncRNAs that were altered in cancer (*Figure 1B*). A final candidate list of 40 lncRNAs that are enriched in ESCs, and dysregulated in cancer, was identified (*Table 1*). Our candidate list contains lncRNAs that have a wide range of expression and also contains several previously identified lncRNAs that have been found to be dysregulated in cancer (*NEAT1*, *FIRRE*, *XIST*, *DANCR*, and *GAS5*), verifying the validity of the approach (*Figure 1—figure supplement 1A*; *Ji et al., 2019*; *Soudyab et al., 2016*; *Yuan et al., 2016*).

We analyzed the ENCODE expression datasets of adult mouse tissue to compare the expression levels of the candidates across tissues (*Figure 1C*). lncRNAs are known to have distinct expression patterns across different tissues, and our results support the notion that lncRNAs are generally not pan-expressed. Interestingly, many of the identified lncRNAs are enriched in embryonic liver, which is the organ with the most regenerative capacity, yet never grows past its original size.

From here, we decided to focus on liver-enriched candidate mouse lncRNAs, especially those that were primarily dysregulated in liver cancers. Because HCC is one of the deadliest cancers and has inadequate treatment options, we focused on lncRNAs that were dysregulated in HCC, *LINC00862*, *TSPOAP-AS1*, *MIR17HG*, and *SNHG5*, with their mouse counterparts being *Gm19705*, *Mir142hg*, *Mir17hg*, and *Snhg5*, respectively. Out of these four lncRNAs that were detected to be amplified in HCC, *LINC00862* was the highest at 13% of all liver cancer cases (*Figure 1—figure supplement 1B*). We assayed *LINC00862* expression in human samples obtained from healthy and cirrhotic livers and HCC nodules. Indeed, we found that levels of *LINC00862* were elevated in HCC tumor nodules, but also in cirrhotic liver, suggesting that it may play a role in HCC progression (*Figure 1D*). In addition, we also assayed *LINC00862* expression in human HCC cell lines and found it to be upregulated in numerous HCC cell lines compared to the normal human liver cell line, THLE-2 (*Figure 1E*).

In order to use a more tractable model system, we assessed the conservation of *LINC00862* and its potential mouse counterpart, *GM19705*, which was internally designated as *lnc05* in previous analyses (*Bergmann et al., 2015*). While much shorter, *GM19705* has 51% sequence identity and the gene order is syntenically conserved, although a reversal event most likely occurred within the locus (*Figure 1—figure supplement 1C*). Weighted gene correlation network analysis of *GM19705* identified that its expression is highly correlated with those of cell cycle genes, such as *BRCA1* and *BRCA2* (*Figure 1—figure supplement 1D*). GO term analysis of the cluster identified cell cycle processes as highly enriched, indicating that *GM19705* may play a role in the regulation of the cell cycle (*Figure 1—figure supplement 1E*). Reanalysis of previously published single-cell analysis of normal adult mouse liver (*Tabula Muris Consortium et al., 2018*) identified *GM19705* expression to be low overall, as expected, but highly expressed exclusively in a subset of hepatocytes (*Figure 1—figure supplement 1F*).

Our analysis identified *GM19705/LINC00862* as a lncRNA that is expressed in ESCs and dysregulated in HCC. We found that *GM19705* is also highly expressed in developing liver and exclusively in adult hepatocytes, and it may have a potential function to regulate the cell cycle. Therefore, we named this mouse lncRNA – <u>P</u>luripotency and <u>H</u>epatocyte <u>A</u>ssociated <u>R</u>NA <u>O</u>verexpressed in <u>H</u>CC, or *PHAROH*.

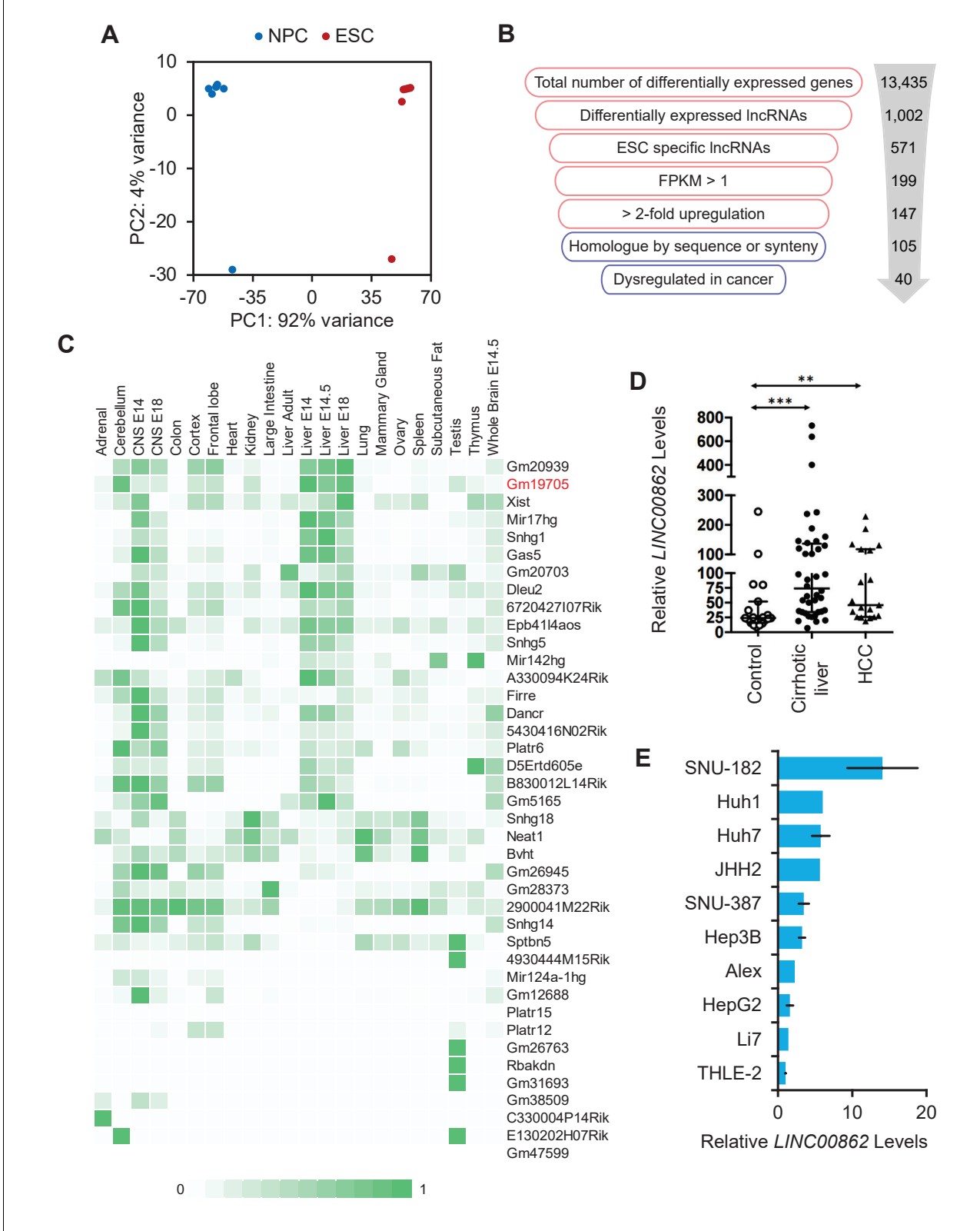

**Figure 1.** Long non-coding RNA (lncRNA) screen to identify transcripts enriched in embryonic stem cells (ESCs) and dysregulated in cancer. (**A**) Principal component analysis (PCA) plot of 10 RNA-seq libraries from mouse-derived ESCs, and two from cell lines. Differentiation from ESCs to neural progenitor cells (NPCs) created the largest difference in variance, while there was minimal difference between isolated clones vs. cell lines. (**B**) Workflow of the filtering process performed to obtain ESC-enriched lncRNAs that are also dysregulated in cancer. Red indicates analysis performed in mouse and

*Figure 1 continued on next page*

*Figure 1 continued*

blue indicates human. (C) lncRNA candidate expression across ENCODE tissue datasets show that lncRNAs are mostly not pan-expressed, but are rather tissue specific. Counts are scaled per row. (D) *LINC00862* is upregulated in both human cirrhotic liver and hepatocellular carcinoma (HCC) tumor samples when compared to control patient liver tissue samples. **p<0.01; ***p<0.005; Student's t-test. (E) *LINC00862* is upregulated in various human HCC cell lines.

The online version of this article includes the following figure supplement(s) for figure 1:

**Figure supplement 1.** LncRNA screen to identify transcripts enriched in ESCs and dysregulated in cancer, related to *Figure 1*.

## *PHAROH* is a novel lncRNA that is highly expressed in embryonic liver and mouse HCC

*PHAROH* is an intergenic lncRNA located on mouse chr1:1qE4. 5′ and 3′ rapid extension of cDNA ends (RACE) revealed the presence of two isoforms that share two common exons and are both ~450 nt (*Figure 2A*). In silico analysis of the coding potential by three independent algorithms, which use codon bias (CPAT/CPC) and comparative genomics (PhyloCSF), all point towards the low coding potential score of *PHAROH*, compared to the *Gapdh* control (*Figure 2—figure supplement 1A, B*). From here on, only qPCR primers that amplify common exons were used. We confirmed expression levels of *PHAROH* in developing liver by assaying the liver bud from E14 and E18 embryos and found that they were seven- to ninefold enriched compared to adult liver (*Figure 2B*). Because the liver is one of the main sites of hematopoiesis in the embryo, we measured *PHAROH* levels in embryonic blood and found that expression was exclusive to the liver, and not to hematopoietic cells (*Figure 2—figure supplement 1C*). *PHAROH* was also found to be upregulated in a partial hepatectomy (PH) model of liver regeneration (*Figure 2—figure supplement 1D*), where the expression was correlated with time points of concerted DNA synthesis, but did not fluctuate across the cell cycle (*Figure 2—figure supplement 1E*). To confirm *PHAROH*'s involvement in HCC, we used a diethylnitrosamine (DEN)-induced carcinogenic model of liver injury. By 11 months post DEN treatment, we were able to visualize HCC tumor nodules, which had elevated levels of *PHAROH* (*Figure 2C*). In order to facilitate the molecular and biochemical study of *PHAROH*, we chose two mouse HCC cell lines, Hepa1-6 and Hepa1c1c7, and indeed found that *PHAROH* was 3- to 4-fold more enriched than in ESCs, and 8- to 10-fold increased over the AML12 mouse normal hepatocyte cell line (*Figure 2D*).

Single-molecule RNA-FISH revealed that *PHAROH* is entirely nuclear in ESCs, with an average of 3–5 foci per cell, whereas it is evenly distributed between the nucleus and cytoplasm in Hepa1-6 cells, with an average of 25 foci per cell (*Figure 2E, F*). Isoform 1 is expressed mostly in ESCs while isoform 2 of *PHAROH* dominates HCC cell lines (*Figure 2A*, *Figure 2—figure supplement 1F*). Cellular fractionation of Hepa1-6 cells corroborates the RNA-FISH-determined localization of *PHAROH* as well, which *GAPDH* and *MALAT1* localized correctly to previously determined cellular fractions (*Figure 2G*). Additional lncRNAs tested, such as *XIST*, *FIRRE*, and *NEAT1*, also localized to their expected cellular fractions (*Figure 2—figure supplement 1G*). It was also determined that *PHAROH* has a relatively longer half-life in the Hepa1-6 cell line (10.8 hr) compared to *MALAT1* (8.0 hr) and *XIST* (4.2 hr) (*Figure 2—figure supplement 1H*; *Tani et al., 2012*; *Yamada et al., 2015*). Taken together, *PHAROH* is an ESC and fetal liver-specific lncRNA that is upregulated in the context of HCC.

## Targeted knockout of *PHAROH*

To evaluate the functional role of *PHAROH*, we generated targeted knockouts using CRISPR/Cas9 technology. Two sgRNA guides were designed to delete a region ~700 bp upstream of the TSS, and ~100 bp downstream of the TSS. We chose to transiently express enhanced specificity Cas9 (eSpCas9-1.1) in order to increase specificity, decrease off-target double-stranded breaks, and also to avoid stable integration of Cas9 endonuclease due to its transformative potential (*Slaymaker et al., 2016*). In addition to using two guides targeting *PHAROH*, we used an sgRNA targeting Renilla luciferase as a non-targeting control. Each guide was cloned into a separate fluorescent protein vector (GFP or mCherry) to allow for subsequent selection. Cells were single-cell sorted 48 hr after nucleofection to account for heterogeneity of deletions among a pooled cell population, which may give certain cells a growth advantage. 85% of the cells were GFP+/mCherry+, and we

**Table 1.** 40 LncRNAs that are enriched in ESCs and dysregulated in cancer.

| Gene name | Sequence homology | Synteny | Human homologue |
|---|---|---|---|
| Platr15 | - | + | LOC284798 |
| 4930444M15Rik | 64.4% of bases, 99.9% of span | + | In TUSC8 region |
| 5430416N02Rik | 16.6% of bases, 100.0% of span | + | Thap9-AS1 |
| Platr6 | 45.2% of bases, 85.5% of span | + | LINC01010 |
| 6720427I07Rik | 94.3% of bases, 100.0% of span | + | LINC02603 |
| B830012L14Rik | 57.4% of bases, 83.8% of span | + | Meg8 (GM26945) |
| C330004P14Rik | - | + | LINC01625 |
| Gm38509 | 22.9% of bases, 84.4% of span | + | LINC01206 |
| A330094K24Rik | 54.7% of bases, 100.0% of span | + | C18orf25 (PCG) |
| Bvht | 53.2% of bases, 100.0% of span | + | Carmn |
| Dancr | 48.2% of bases, 49.0% of span | + | Dancr |
| 2900041M22Rik | 50.2% of bases, 60.5% of span | + | LINC01973 |
| Dleu2 | 72.8% of bases, 100.0% of span | + | Dleu2 |
| E130202H07Rik | 61.7% of bases, 65.2% of span | | Tusc8 |
| Epb41l4aos | 69.0% of bases, 100.0% of span | + | Epb41l4a-AS1 |
| Firre | 7.0% of bases, 14.5% of span | + | Firre |
| Gm20939 | - | + | LINC00470 |
| Gas5 | 71.3% of bases, 97.7% of span | + | Gas5 |
| Gm12688 | 92.6% of bases, 100.0% of span | + | FOXD3-AS1 |
| Gm47599 | 21.6% of bases, 85.0% of span | + | Socs2-AS1 |
| Gm19705 | 27.6% of bases, 47.8% of span | + | LINC00862 |
| Gm20703 | 79.2% of bases, 100.0% of span | + | GAPLINC |
| Gm26763 | 3.6% of bases, 3.8% of span | + | Smarca5-AS1 |
| Gm26945 | 65.4% of bases, 67.8% of span | + | Meg8 |
| AC129328.1 | - | + | LINC01340, |
| Gm28373 | 44.6% of bases, 83.5% of span | + | Itpk1-AS1 |
| Gm31693 | 12.7% of bases, 24.9% of span | + | LINC00578 |
| Mir124a-1hg | 91.7% of bases, 100.0% of span | + | LINC00599 |
| Mir142hg | 74.5% of bases, 100.0% of span | + | TSPOAP1-AS1 |
| Mir17hg | 74.7% of bases, 100.0% of span | + | Mir17Hg |
| Neat1 | 37.5% of bases, 100.0% of span | + | NEAT1 |
| Platr12 | 16.2% of bases, 33.7% of span | + | GPR1-AS |
| Rbakdn | 96.4% of bases, 99.1% of span | + | Rbakdn |
| Snhg1 | 73.3% of bases, 89.2% of span | + | Snhg1 |
| Snhg14 | 4.5% of bases, 5.4% of span | + | Snhg14 |
| D5Ertd605e | - | + | Pan3-AS1 |
| Snhg18 | 83.3% of bases, 100.0% of span | + | Snhg18 |
| Snhg5 | 67.8% of bases, 81.6% of span | + | Snhg5 |
| Sptbn5 | 78.8% of bases, 100.0% of span | + | Sptbn5 |
| Xist | 70.1% of bases, 100.0% of span | + | Xist |

selected four clones for subsequent analysis (*Figure 3—figure supplement 1A*). All selected clones had the correct homozygous deletion when assayed by genomic PCR (*Figure 3A*). qRT-PCR indicated that *PHAROH* was knocked down 80–95% (*Figure 3B*).

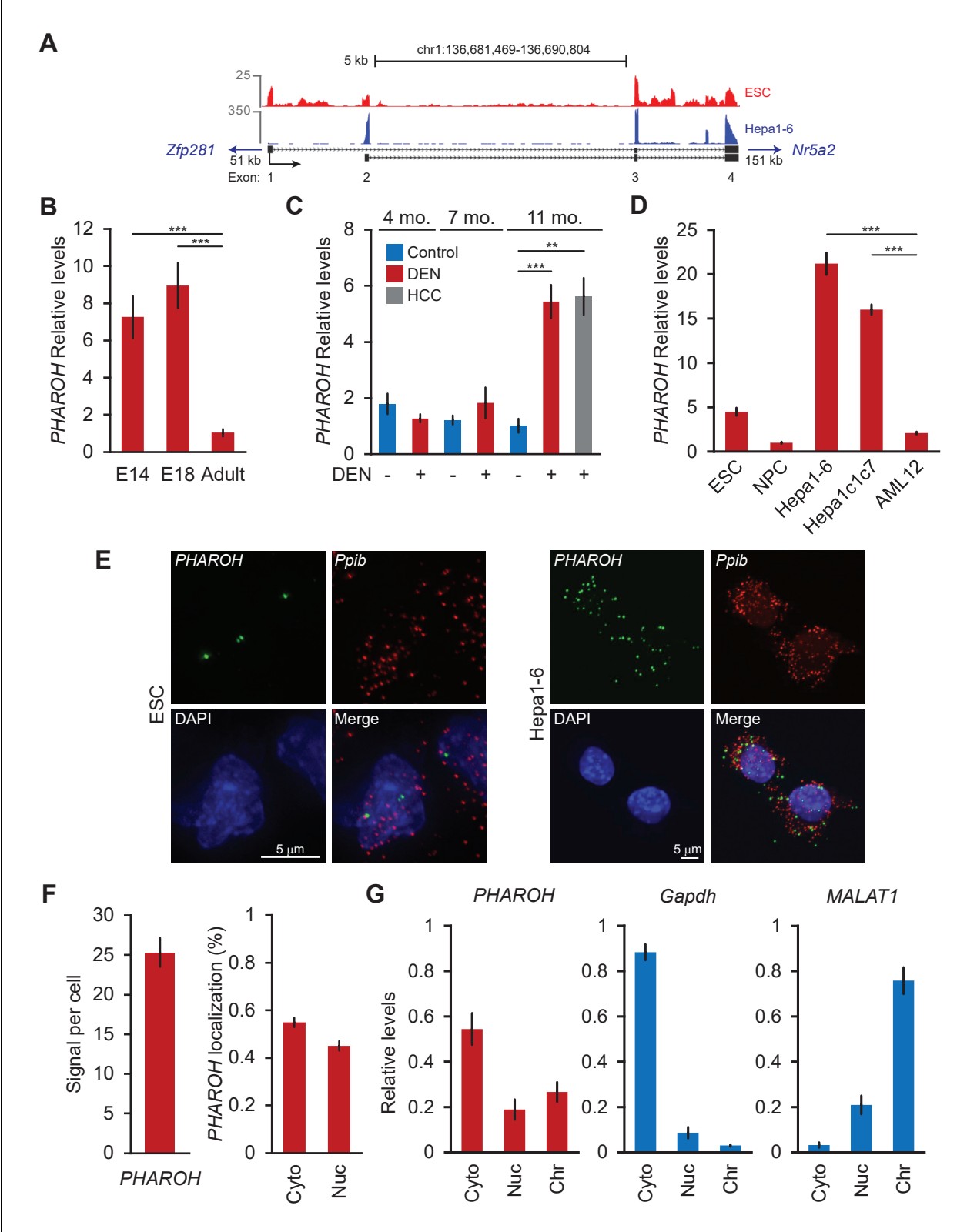

**Figure 2.** *PHAROH* long non-coding RNA (lncRNA) is highly expressed in embryonic stem cells (ESCs), embryonic liver, models of hepatocarcinogenesis, and hepatocellular carcinoma (HCC) cell lines. (**A**) 5′ 3′ rapid extension of cDNA ends (RACE) reveals two isoforms for *PHAROH*, which have exons 3 and 4 in common. *PHAROH* is an intergenic lncRNA where the nearest upstream gene is *Zfp218* (51 kb away), and downstream is *Nr5a2* (151 kb away). RNA-seq tracks of ESC (red) and Hepa1-6 (blue) cells show cell-type-specific isoform expression of *PHAROH*. (**B**) *PHAROH* is highly
*Figure 2 continued on next page*

*Figure 2 continued*

expressed in embryonic liver in E14 and E18 mice, but not adult liver (**p<0.01; ***p<0.005; Student's t-test). (**C**) A diethylnitrosamine (DEN) model of hepatocarcinogenesis shows high upregulation of *PHAROH* in the liver and HCC tumor nodules (gray bar) in DEN-treated mice (**p<0.01; ***p<0.005; Student's t-test). (**D**) *PHAROH* is upregulated in HCC cell lines (Hepa1-6 and Hepa1c1c7) compared to normal mouse hepatocytes (AML12) (***p<0.005; Student's t-test). (**E**) Single-molecule RNA-FISH of *PHAROH* in ESCs shows nuclear localization and an average of 3–5 foci per cell. In Hepa1-6 cells, *PHAROH* shows 25 foci per cell, distributed evenly between the nucleus and cytoplasm (n = 75 cells for each sample). *Ppib* is used as a housekeeping protein coding gene control. (**F**) Quantitation of panel *PHAROH* foci in panel (**E**) in HepA1-6 cells. (**G**) Cellular fractionation of Hepa1-6 cells shows equal distribution of *PHAROH* in the cytoplasm and nucleus, where it also binds to chromatin. *Gapdh* is predominantly cytoplasmic, and *MALAT1* is bound to chromatin.

The online version of this article includes the following figure supplement(s) for figure 2:

**Figure supplement 1.** PHAROHlncRNA is highly expressed in ESCs, embryonic liver, models of hepatocarcinogenesis, and HCC cell lines, related to *Figure 2*.

We assayed the proliferative state of the *PHAROH* knockout clones and found a decrease in proliferation. The doubling time of the knockout clones increased to 18.2 hr compared to the wildtype doubling time of 14.8 hr, and ectopic expression of PHAROH reduced the doubling time to nearly wildtype levels (*Figure 3C*). Ectopic expression of *PHAROH* also successfully rescued the proliferation phenotype in the knockout clones, suggesting that *PHAROH* functions in trans (*Figure 3D*). Migration distance was also decreased by 50% in the knockout clones (*Figure 3E*).

In addition to assessing the role of *PHAROH* in knockout clones, we also employed the use of antisense oligonucleotides (ASO) to knockdown *PHAROH*. We treated cells independently with a control scrambled cEt ASO or two independent cEt ASOs complementary to the last exon of *PHAROH*. ASOs were nucleofected at a concentration of 2 µM, and we are able to achieve a >90% knockdown at 24 hr, and an ~50% knockdown was still achieved after 96 hr (*Figure 3—figure supplement 1B*). Proliferation assays using manual cell counts and MTS assay show a 50% reduction in proliferation at 4 days (96 hr), similar to that achieved in our knockout clones, further supporting a role of *PHAROH* in cell proliferation (*Figure 3—figure supplement 1C*). Addition of the ASO into the medium allowed for the knockdown to persist for longer duration to study the impact on clonogenic ability (*Figure 3F*). Colony formation assays demonstrated that knockdown of *PHAROH* significantly inhibits clonogenic growth of HCC cells in a dose-dependent manner (*Figure 3G*, *Figure 3—figure supplement 1D*).

To investigate the global effect of *PHAROH* depletion, we performed poly(A)+RNA seq on control and knockout clones (*Figure 4—figure supplement 1A, B*). We identified 810 differentially expressed genes, and GO term analysis revealed regulation of cell proliferation, locomotion, and cell motility as the highest enriched terms (*Figure 4A*). To determine if these differentially expressed genes were predominantly controlled by common transcription factors, we performed de novo and known motif analysis. Interestingly, promoter motif analysis of differentially expressed genes revealed enrichment of the Myc motif in our dataset, suggesting a subset of the genes were under the transcriptional control of Myc (*Figure 4—figure supplement 1C*). This was intriguing because Myc is known to regulate cell proliferation and is highly amplified in nearly half of HCCs (*Zheng et al., 2017*). However, Myc expression changes were not detected in our RNA-seq analysis, nor was there any statistically significant change compared to sgRenilla controls when assayed by qRT-PCR (*Figure 4B*). Strikingly, *MYC* protein levels were substantially decreased in all of the *PHAROH* knockout clones, as detected by western blot and immunofluorescence, suggesting that *PHAROH* regulates *Myc* post-transcriptionally (*Figure 4C*, *Figure 4—figure supplement 1D, E*). qRT-PCR of genes downstream of Myc that were identified through our analysis were also significantly downregulated in *PHAROH* knockout clones (*Figure 4D*). Thus, we suggest that depletion of *PHAROH* decreases *MYC* protein levels and ultimately cell proliferation.

## RAP-MS identifies TIAR as the major interactor of *PHAROH*

lncRNAs can act as structural scaffolds to promote interaction between protein complexes or to sequester a specific protein (*Lee et al., 2016*; *Tsai et al., 2010*). Because modulation of *PHAROH* levels change *Myc* protein levels, but not mRNA levels to a significant degree, we hypothesized that *PHAROH* may be regulating the translation of *MYC* through a protein mediator. In order to search for *PHAROH* interacting proteins, we used a pulldown method adapted from the previously

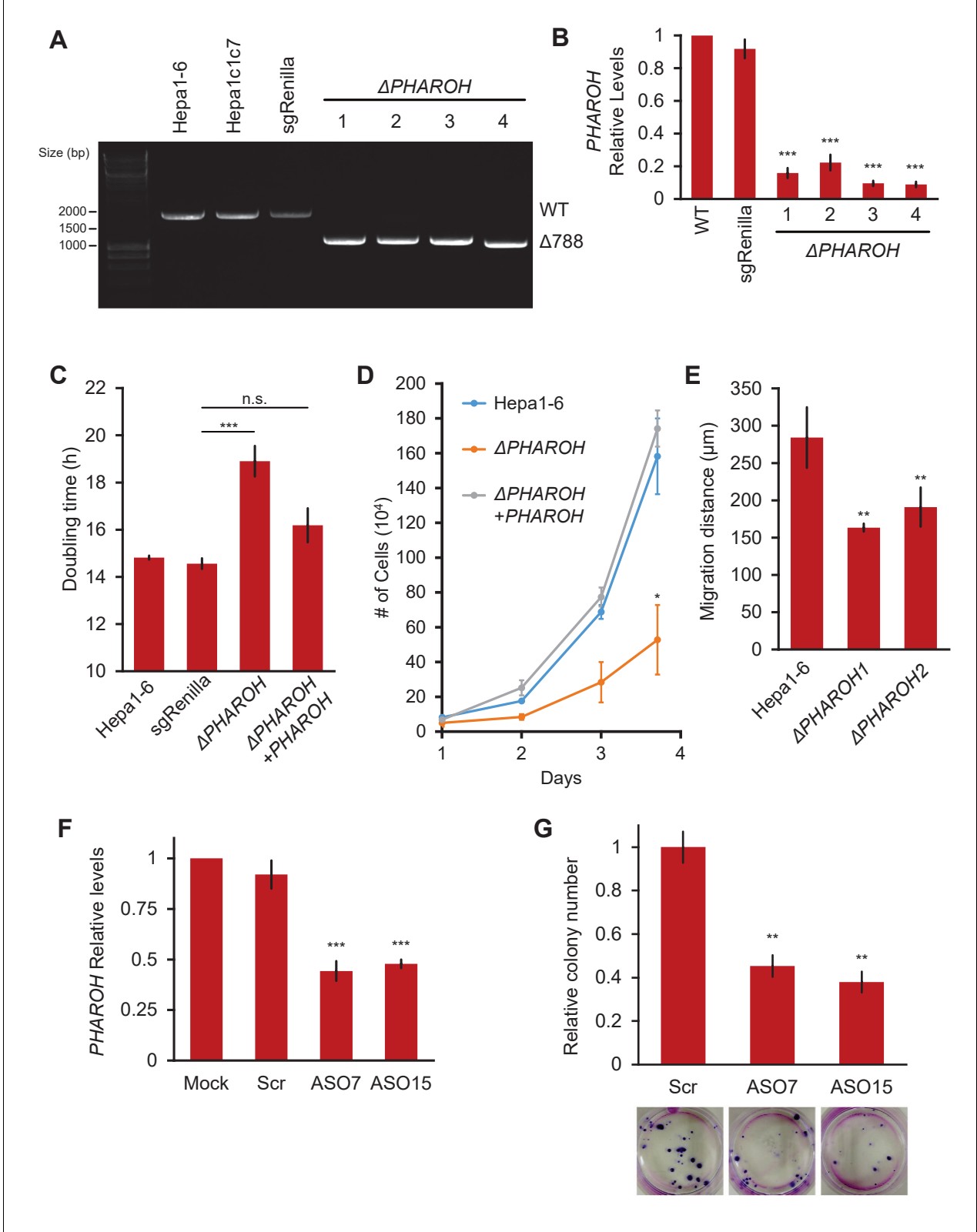

**Figure 3.** Depletion of *PHAROH* results in a proliferation defect. (**A**) Four isolated clones all have a comparable deletion of 788 bp. The wildtype band is ~1.8 kb. (**B**) qRT-PCR of *PHAROH* knockout clones show a >80% reduction in *PHAROH* levels (\*\*\*p<0.005; Student's t-test). (**C**) Aggregated doubling time of clones shows knockout of *PHAROH* increases doubling time from 14.8 hr to 18.6 hr. Addition of *PHAROH* back into knockouts rescues this defect (\*\*\*p<0.005; Student's t-test). (**D**) Manual cell counting shows proliferation defect in *PHAROH* knockout cells that is rescued by ectopic
*Figure 3 continued on next page*

*Figure 3 continued*

expression of *PHAROH* (*p<0.05; Student's t-test). (E) Migration distance for *PHAROH* knockout clones is decreased by 50% (**p<0.01; Student's t-test). (F) 50% knockdown of *PHAROH* can be achieved using both antisense oligonucleotide (ASO)7 and ASO15 at 24 hr (***p<0.005; Student's t-test). (G) Colony formation assay of Hepa1-6 cells that are treated with scrambled or *PHAROH* targeting ASOs. After seeding 200 cells and 2 weeks of growth, a 50% reduction in relative colony number is observed (**p<0.01; Student's t-test).

The online version of this article includes the following figure supplement(s) for figure 3:

**Figure supplement 1.** Depletion of PHAROH results in a proliferation defect, related to *Figure 3*.

published RNA antisense purification-mass spectrometry (RAP-MS) (*McHugh et al., 2015*). In lieu of pooling all available antisense capture biotinylated oligonucleotides (oligos), we reasoned that individual oligos may be similarly effective and can be used as powerful biological replicates. In addition, we would minimize oligo-specific off targets by verifying our results with multiple oligos. To this end, we screened through five 20-mer 3′ biotinylated DNA oligos that tiled the length of *PHAROH* and found that four out of the five oligos pulled down >80% of endogenous *PHAROH*, while the pull-down of a control RNA, *PPIB*, remained low (*Figure 5A*, *Figure 5—figure supplement 1A*).

For elution of *PHAROH*, we tested a range of temperatures and found that the elution efficiency reaches the maximum at 40℃, and thus we used this temperature for further experiments (*Figure 5B*). The remaining level of *PHAROH* RNA on the beads was the direct inverse of the eluate (*Figure 5—figure supplement 1B*). We chose *PPIB* as a negative control because it is a housekeeping mRNA that is expressed on the same order of magnitude as *PHAROH* and is not expected to interact with the same proteins. We screened through 10 oligos against *PPIB* and found only one that pulled *PPIB* down at ~60% efficiency, and eluted at the same temperature as *PHAROH* (*Figure 5—figure supplement 1C, D*). Off-target RNA pulldown, such as *PHAROH* and *18S* rRNA, remained minimal when using the oligo antisense to *PPIB* (*Figure 5—figure supplement 1C*).

To identify proteins that bind to *PHAROH*, we analyzed two independent oligos that target *PHAROH* and two replicates of *PPIB* on a single 4-plex iTRAQ (isobaric tag for relative and absolute quantitation) mass spectrometry cassette and identified a total of 690 proteins. By plotting the $\log_2$ enrichment ratio of *PHAROH* hits divided by *PPIB* hits, quadrant I will contain proteins that both oligos against *PHAROH* recognize, and quadrant III will be enriched for proteins that bind specifically to *PPIB*. Quadrant III was enriched for keratins, elongation factors, and ribosomal proteins. Interestingly, the top hit in quadrant I is nucleolysin TIAR (TIAL-1), an RNA-binding protein that controls mRNA translation by binding to AU-rich elements in the 3′ UTR of mRNA (*Figure 5C*, *Table 2*; *Mazan-Mamczarz et al., 2006*). TIAR is present in <10% of all experiments queried on Crapome.org (31/411). Immunoblots for TIAR confirm the mass spectrometry data in that TIAR is specific to *PHAROH* pulldown oligos and also is eluted at 40℃ (*Figure 5D*). Additional controls that are not complementary to the mouse genome and oligos targeting *PHAROH* also confirm the TIAR hit, and it is reproducible in two independent HCC cell lines (*Figure 5E*). RNase A treatment of the lysate largely abolished the interaction, which indicates that the interaction is RNA mediated, and not the result of direct binding to the oligo (*Figure 5E*). Immunoprecipitation of TIAR and subsequent extraction of interacting RNA shows enrichment for *PHAROH* when compared to *PPIB* and IgG control (*Figure 5F*). Thus, together these data indicate that TIAR is a bona fide interactor of *PHAROH*.

## A 71-nt sequence in *PHAROH* has four TIAR binding sites

A previous study on TIAR has mapped its RNA recognition motif (RRM) across the transcriptome (*Meyer et al., 2018*). Analysis of *PHAROH*'s sequence reveals that TIAR binding sites are enriched in the 5′ end of the transcript of both isoforms (*Figure 6A*). To determine if there is any conserved structure within *PHAROH* that mediates this interaction, RNA folding prediction algorithms, mFold and RNAfold, were used. The two strongest TIAR binding sequences (TTTT and ATTT/TTTA) were mapped onto 10 outputted predicted structures (*Figure 6—figure supplement 1A*). Strikingly, four out of the seven binding sites consistently mapped to a hairpin that was conserved throughout all predicted structures. Three of the strongest binding motifs localize to the stem of the hairpin, while one secondary motif resides in a bulge (*Figure 6B*). These data indicate that the sequence is a highly concentrated site for TIAR binding and is designed to potentially sequester multiple copies of TIAR.

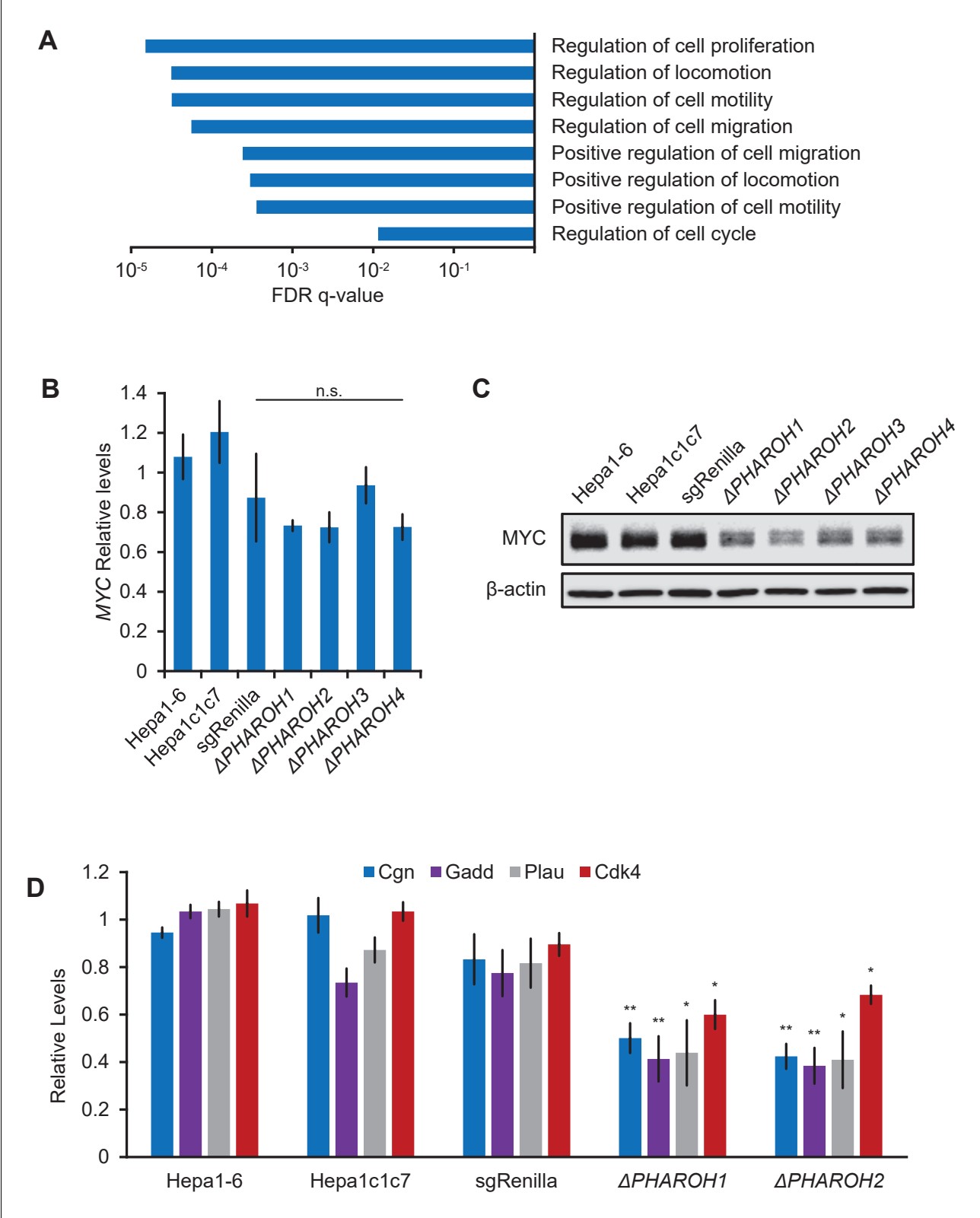

**Figure 4.** Gene expression analysis of *PHAROH* knockout cells reveals a link to MYC. (**A**) GO term analysis of differentially expressed genes shows enrichment of cell proliferation and migration genes. (**B**) qRT-PCR of Myc mRNA levels indicates that Myc transcript does not appreciably change when *PHAROH* is knocked out. (**C**) Western blot analysis of MYC protein shows downregulation of protein levels in *PHAROH* knockout cells. β-Actin is used

*Figure 4 continued on next page*

*Figure 4 continued*

as a loading control. (D) qRT-PCR of genes downstream of Myc shows a statistically significant decrease in expression (*p<0.05; **p<0.01; Student's t-test).

The online version of this article includes the following figure supplement(s) for figure 4:

**Figure supplement 1.** Gene expression analysis ofPHAROHknockout cells reveals a link to MYC, related to *Figure 4*.

RNA electromobility shift assay (EMSA) of the hairpin and recombinant human TIAR showed that as TIAR concentration increases it binds to the *PHAROH* hairpin multiple times (*Figure 6C*). TIAR has a preference to bind two and four times, rather than once or three times. Densitometry quantification of the remaining free probe shows that TIAR has an approximate dissociation constant of 2 nM, consistent with the literature (*Kim et al., 2011*; *Figure 6—figure supplement 1B*). Addition of an antibody against TIAR creates a supershift, showing that the interaction is specific, while addition of IgG does not. The interaction can be abolished with addition of 20× unlabeled probe as well (*Figure 6E*, left panel).

To determine if binding of TIAR is specific to the sequence and mapped motifs, we created sequential mutations of the hairpin by changing the non-canonical Watson–Crick base pairs (starred and in red) to canonical ones (*Figure 6B*). Mutation of the first binding site (m1) slightly reduced specificity of TIAR to the hairpin, but changes the preference of TIAR binding to one and two units (*Figure 6E*, right panel). Mutation of m2 greatly reduced TIAR association, and only two bands are highly visible (*Figure 6E*, right panel). However, mutation of three binding sites (m3) did not appreciably change the pattern, as compared to m2, perhaps suggesting that the weaker binding site is only used cooperatively (*Figure 6—figure supplement 1C*). Mutation of all four binding sites (m4) showed minimal TIAR binding (*Figure 6E*). Taken together, these data indicate that TIAR binds directly to the 71-nt sequence on *PHAROH*, which can fold into a hairpin, and preferentially binds two or four times.

### *PHAROH* modulates Myc translation by sequestering TIAR

TIAR has been shown to bind to the 3′ UTR of mRNAs containing AU-rich elements in order to inhibit their translation (*Mazan-Mamczarz et al., 2006*). It has also been shown that TIAR binds to the 3′ UTR of *Myc* mRNA (*Liao et al., 2007*). Our data suggests that *PHAROH* serves to competitively sequester TIAR in order to allow for increased *MYC* translation. Thus, knockout or knockdown of *PHAROH* will free additional TIAR molecules to bind to the 3′ UTR of *Myc* and inhibit its translation.

We began by determining where TIAR binds to *Myc* mRNA. Mapping PAR-CLIP reads from *Meyer et al., 2018* shows two distinct binding sequences on the human *MYC* mRNA, but only one sequence maps to the mouse genome. The stretch of 53-nt sequence has three distinct regions that are enriched in poly-uridines, but structural prediction largely places the sequences in a loop formation (*Figure 7—figure supplement 1A, B*). RNA EMSA of the 53-nt 3′ UTR and recombinant TIAR showed preference for a singular binding event, and three events are only seen when the binding reaction is saturated by TIAR (*Figure 7A*). ASO-mediated knockdown of *PHAROH* shows reduction of MYC protein similar to the knockouts, but no change in mRNA levels, or TIAR protein levels (*Figure 7B*, *Figure 7—figure supplement 1C*). While mRNAs are generally much more highly expressed than lncRNAs, *Myc* is only threefold more expressed than *PHAROH* in HCC cell lines (*Figure 7B*). In addition, there are multiple TIAR binding sites on *PHAROH*, which increases the feasibility of a competition model (*Figure 7B*).

Next, we tested this hypothesis in vitro by allowing TIAR to bind to the 53-nt Myc 3′ UTR and titrating increasing amounts of *PHAROH* or the mutant *PHAROH* transcript. The wildtype *PHAROH* hairpin can be seen to compete with *Myc* very effectively at nearly all tested ratios, with near complete competition at 10:1 ratio (*Figure 7C*). However, the fully mutant *PHAROH* was not able to compete with *Myc* nearly as effectively and was only seen to be slightly effective at the 10:1 ratio (*Figure 7C*). This data suggests that the *PHAROH* has the capability to successfully compete with the *Myc* 3′ UTR binding site in a sequence-dependent manner.

In addition, we cloned the full-length *Myc* 3′ UTR into a dual luciferase reporter construct in order to test our hypothesis in cells. We found that addition of *PHAROH* does indeed increase the

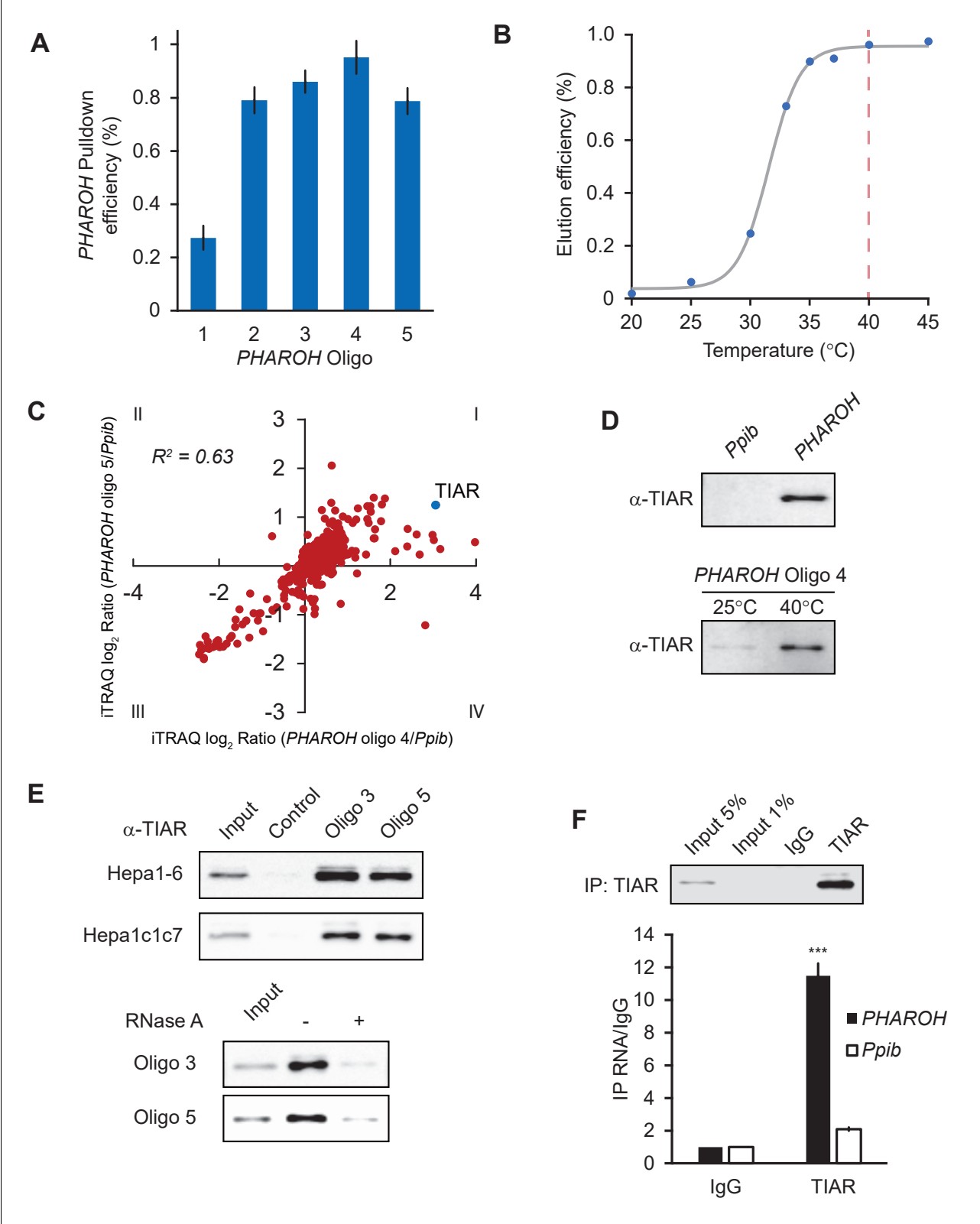

**Figure 5.** RNA antisense purification-mass spectrometry (RAP-MS) identifies TIAR as a major interactor of *PHAROH*. (**A**) Five different biotinylated oligos antisense to *PHAROH* were screened for pulldown efficiency. Oligos 2–5 can pull down *PHAROH* at ~80% efficiency or greater. (**B**) *PHAROH* can be eluted at a specific temperature. Maximum elution is reached at 40°C. (**C**) iTRAQ results using two different oligos targeting *PHAROH* compared to PPIB reveal nucleolysin TIAR as the top hit. (**D**) TIAR is pulled down by *PHAROH* oligos and is specifically eluted at 40°C, but not by PPIB oligos. (**E**) *Figure 5 continued on next page*

*Figure 5 continued*

TIAR can be pulled down using additional oligos and in two different cell lines. RNase A treatment of the protein lysate diminishes TIAR binding to *PHAROH*, indicating that the interaction is RNA-dependent. (F) Immunoprecipitation of TIAR enriches for *PHAROH* transcript when compared to IgG and PPIB control (\*\*\*p<0.005; Student's t-test).

The online version of this article includes the following figure supplement(s) for figure 5:

**Figure supplement 1.** RNA antisense purification-mass spectrometry (RAP-MS) identifies TIAR as a major interactor of *PHAROH*, related to *Figure 5*.

luciferase signal by ~50% in a dose-dependent manner while the mutant *PHAROH* did not (*Figure 7D*, *Figure 7—figure supplement 1D*).

Given that the knockdown or knockout of *PHAROH* reduces MYC levels due to the release of TIAR, we asked whether MYC protein levels would change in the context of *PHAROH* overexpression. Compared to GFP transfection, overexpression of *PHAROH* increases *MYC* protein levels; however, overexpression of mutant *PHAROH* did not change the protein levels of MYC (*Figure 7E*). Modulation of *PHAROH* or TIAR levels did not have an effect on *Myc* mRNA levels (*Figure 7—figure supplement 1E*).

## Discussion

Studies of the transcriptome have shed important insights into the potential role of the non-coding RNA portion of the genome in basic biology as well as disease. As such, lncRNAs can serve as biomarkers, tumor suppressors, or oncogenes, and have great potential as therapeutic targets (reviewed in *Arun et al., 2018*). Here, we identified a lncRNA, *PHAROH*, that is upregulated in mouse ESCs, embryonic and regenerating adult liver, and in HCC. It also has a conserved human ortholog, which is upregulated in human patient samples from cirrhotic liver and HCC. Genetic knockout or ASO knockdown of *PHAROH* results in a reduction of cell proliferation, migration, and colony formation.

To elucidate the molecular mechanism through which *PHAROH* acts in proliferation, we used RNA-seq and mass spectrometry to provide evidence that *PHAROH* regulates MYC translation via sequestering the translational repressor TIAR in trans. Modulation of *PHAROH* levels reveals that it is positively correlated with MYC protein level, which is well known to be associated with HCC and is amplified in nearly 50% of HCC tumors (*Peng et al., 1993*). In addition, MYC has been characterized as a critical player in liver regeneration (*Zheng et al., 2017*). We identified TIAR as an intermediate player in the *PHAROH*-MYC axis, which has been reported to bind to the 3′ UTR of MYC mRNA and suppress its translation (*Mazan-Mamczarz et al., 2006*). While TIAR is an RNA-binding protein that is known for its role in stress granules (*Kedersha et al., 1999*), we do not detect stress granule formation in our HCC cell lines as assayed by immunofluorescence for TIAR (*Figure 7—figure supplement 1F*). As such, the role of *PHAROH*-TIAR lies outside the context of stress granule function. Interestingly, overexpression of TIAR is a negative prognostic marker for HCC survival (*Figure 7—figure supplement 1G*; *Uhlen et al., 2017*). As the primary mutation of HCC is commonly amplification of MYC, it is possible that TIAR is upregulated in an attempt to curb MYC expression.

Our analysis maps the *PHAROH*-TIAR interaction to predominantly occur at a 71-nt hairpin at the 5′ end of *PHAROH*. While *PHAROH* has two main isoforms that are selectively expressed in ESCs and HCC, the hairpin is commonly expressed in both isoforms. TIAR has been classified as an ARE-binding protein that recognizes U-rich and AU-rich sequences. Kinetic and affinity studies have found that TIAR has a dissociation constant of ~1 nM for U-rich sequences and ~14 μM for AU-rich sequences (*Kim et al., 2011*). One question that is apparent in the RNA-binding protein field is how RBPs acquire their specificity. While there have been studies that analyze target RNA structure or RRM structure, why RBPs bind one transcript over another with a similar sequence is still an open question. For example, the 3′ UTR of Myc contains multiple U-rich stretches, ranging from 3 to 9 resides. It has been reported that TIAR binds efficiently to uridylate residues of 3–11 length, yet PAR-CLIP data only reveals two binding events in the human *MYC* transcript (*Kim et al., 2011*). In addition, the 53-nt fragment that was assayed in this study contained potentially six TIAR binding sites, yet RNA EMSA analysis revealed a preference for a single binding event (*Figure 7A*). One explanation is that *PHAROH*'s hairpin has uniquely spaced TIAR binding sites. Because the absolute affinity of TIAR

**Table 2.** Top protein candidates that interact with PHAROH.

| Protein hit | Ratio |
| --- | --- |
| Tial1 | 2.15559 |
| Hnrnpab | 1.80692 |
| Rbm3 | 1.77037 |
| Hnrnpd | 1.62883 |
| Hnrnpa1 | 1.6283 |
| Ptbp2 | 1.57804 |
| Hnrnpa3 | 1.53035 |
| Caprin1 | 1.50299 |
| Lmna | 1.37542 |
| Fubp3 | 1.34941 |
| Banf1 | 1.34137 |
| Hnrnpa2b1 | 1.33969 |
| H2afj | 1.3213 |
| Lima1 | 1.20909 |
| Nolc1 | 1.20733 |
| Abcb5 | 1.19592 |
| Nup62 | 1.18297 |
| Elavl1 | 1.09477 |
| Ssbp1 | 1.08439 |
| Hist1h2bc | 1.07366 |
| Itgax | 1.00222 |
| Rbm8a | 0.98396 |
| Dhx9 | 0.95827 |
| Smu1 | 0.94938 |
| Cnbp | 0.9225 |
| Nup93 | 0.82199 |
| Lsm3 | 0.79027 |
| Xrcc5 | 0.78242 |
| Med25 | 0.76892 |
| Actc1 | 0.76507 |
| Khsrp | 0.75921 |
| Actb | 0.75109 |
| Nipsnap1 | 0.75014 |
| Pnn | 0.74713 |
| Hba-a1 | 0.74299 |
| Snrpe | 0.74052 |
| Nol11 | 0.73772 |
| Erh | 0.73354 |
| Psmb1 | 0.72391 |
| Efhd2 | 0.71468 |

to U-rich sequences is relatively high, one molecule may sterically block additional binding events. However, if the binding sites are properly spaced, binding events will be ordered and perhaps even

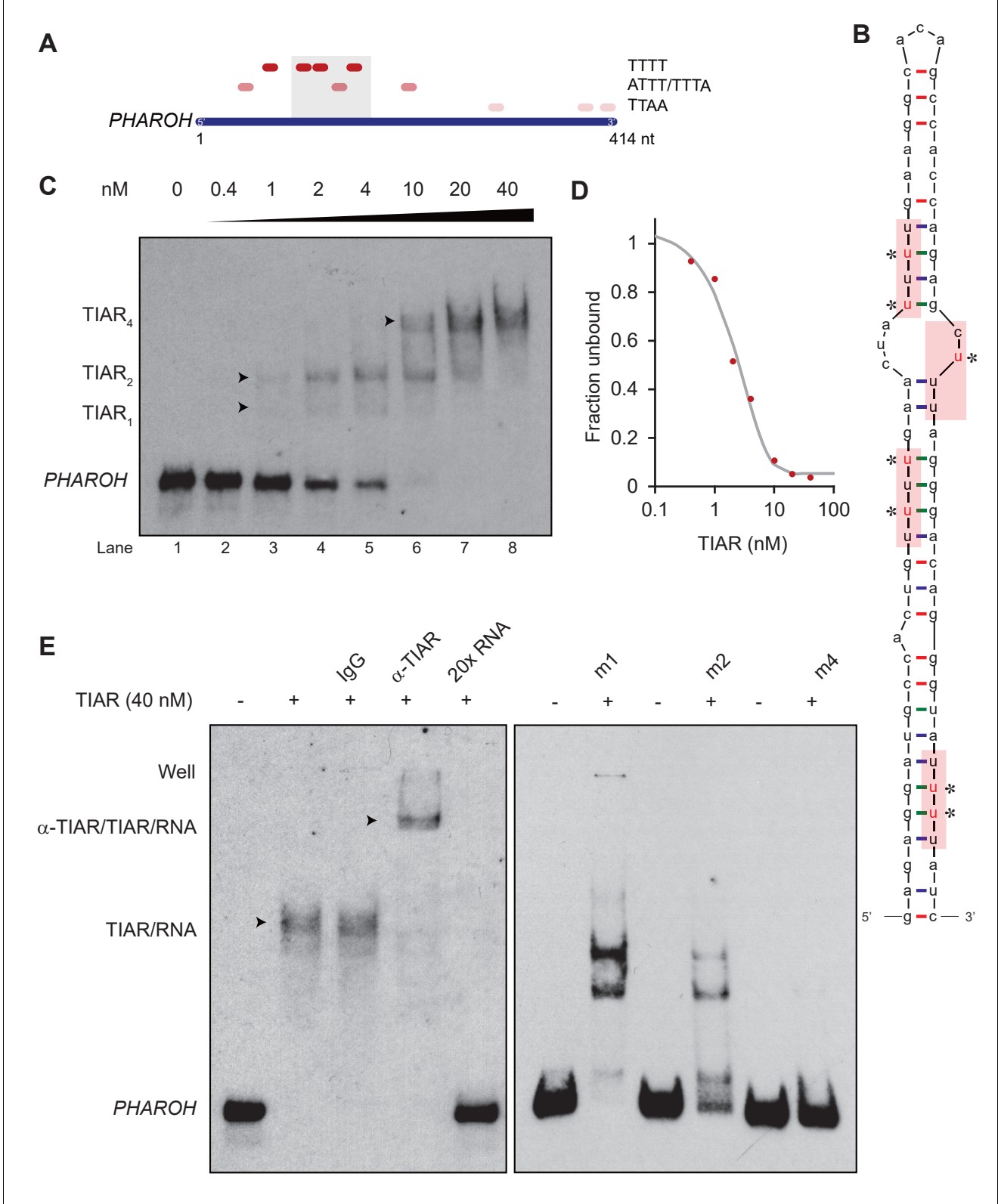

**Figure 6.** TIAR binds to the 5′ end of PHAROH. (**A**) Sequence analysis of *PHAROH* with published TIAR binding motifs shows a preference for the 5′ end of *PHAROH*. (**B**) Schematic of the conserved hairpin of *PHAROH* that contains four potential TIAR binding sites indicated in the red boxes. Mutations created within the *PHAROH* hairpin are indicated in red asterisks. (**C**) RNA electromobility shift assay (EMSA) of the 71-nt *PHAROH* hairpin with human recombinant TIAR shows three sequential shifts as TIAR concentration increases. (**D**) Densitometry analysis of the free unbound probe

*Figure 6 continued on next page*

*Figure 6 continued*

estimates the dissociation constant of TIAR as ~2 nM. (E) TIAR/PHAROH binding is specific as a supershift is created when adding antibody against TIAR, and the interaction can be competed out using 20× unlabeled RNA. RNA EMSA of the mutant hairpins reveals decreasing affinity for TIAR. Mutants were made in a cumulative 5′ to 3′ fashion. M1 shows high signal of single and double occupancy forms, and m2 has reduced signal overall. When all four sites are mutated, binding is nearly abolished.

The online version of this article includes the following figure supplement(s) for figure 6:

**Figure supplement 1.** TIAR binds to the 5′ end of PHAROH, related to *Figure 6*.

cooperative. The average gap between binding sites in the *Myc* fragment is 2 nt, while it is 10 nt in the *PHAROH* hairpin, which allows more flexibility in spacing between each bound protein.

In addition, one aspect that was not explored was the requirement for the formation of the hairpin for TIAR binding. Previous studies used synthesized linear oligos as substrates to test the kinetics of these RBPs, and we also mutated the hairpin in a way such that structure is preserved. TIAR contains three RRMs, which typically recognizes single-stranded RNA. Therefore, binding of TIAR to the 71-nt sequence of *PHAROH* would require unwinding of the potential hairpin, which is energetically unfavorable. It is also known that TIAR's RRM2 mainly mediates ssRNA polyU-binding, but its dsRNA binding capabilities have not been explored (*Kim et al., 2013*). There are examples where multiple RRMs in tandem can allow for higher RNA binding affinity and possibly sandwiching dsRNA, and thus it would be possible that TIAR binding to the multiple sites on the *PHAROH* hairpin is cooperative (*Allain et al., 2000*).

While TIAR may be *PHAROH*'s top interacting protein, it is unknown whether *PHAROH* is TIAR's highest interacting RNA. This would depend on the relative abundances of each RNA species that has the potential to bind TIAR, and TIAR's expression level. This seems to be cell type specific as TIAR was initially studied in immune cells and was shown to predominantly translationally repress *Tnf-α* through binding of the AU-rich sequence in the 3′ UTR (*Piecyk et al., 2000*). In our cell lines, *Tnf-α* is not expressed. Conversely, a screen for proteins that bind to the *Tnf-α* 3′ UTR may not necessarily indicate TIAR as a binder, as evidenced by a recent study (*Ma and Mayr, 2018*). Another recent study had shown that lncRNA *MT1JP* functions as a tumor suppressor and had the capability to bind to TIAR, which suppresses the translation of p53 (*Liu et al., 2016*). However, *MT1JP* is largely cytoplasmic, while TIAR in our context is mainly nuclear. Thus, while TIAR may bind additional mRNAs or lncRNAs, it seems that one of the main targets in HCC cell lines is *Myc*, as supported by statistically significant promoter enrichment of the downstream targets.

In summary, we have identified a lncRNA, *PHAROH*, that is enriched in ESCs and dysregulated in HCC, and found that it acts to sequester TIAR through a hairpin structure in order to regulate *MYC* translation. Additionally, based on synteny and upregulation in human HCC samples, we identified *LINC00862* as the possible human ortholog of PHAROH (*Figure 1D*). Future studies will reveal the therapeutic potential of targeting *PHAROH* to impact liver development/regeneration and HCC.

## Materials and methods

### Cell culture and genomic PCR

All cell culture reagents were obtained from Gibco (Life Technologies), unless stated otherwise. Hepa1-6, Hepa1c1c7, AML12, SNU-182, and THLE-2 cells were obtained from ATCC. Huh7, SNU-387, Hep3B, and HepG2 were generous gifts from Scott Lowe (MSKCC). Hepa1-6, Hepa1c1c7, Huh7, Hep3B, and HepG2 were maintained in DMEM supplemented with 10% FBS and 1% penicillin/streptomycin. SNU-182 and SNU-387 were maintained in RPMI supplemented with 10% FBS and 1% penicillin/streptomycin. AML12 was maintained in DMEM:F12 medium supplemented with 10% fetal bovine serum, 10 μg/ml insulin, 5.5 μg/ml transferrin, 5 ng/ml selenium, and 40 ng/ml dexamethasone. THLE-2 cells were maintained in BEGM (BEGM Bullet Kit; CC3170) where gentamycin/amphotericin and epinephrine were discarded, and extra 5 ng/ml EGF, 70 ng/ml phosphoethanolamine, and 10% fetal bovine serum were added in addition to the supplied additives. ESCs and NPCs were cultured as in *Bergmann et al., 2015*. All cells were cultured in a humidified incubator at 37°C and 5% $CO_2$. Half-life of RNA was determined by adding α-amanitin to a final concentration of

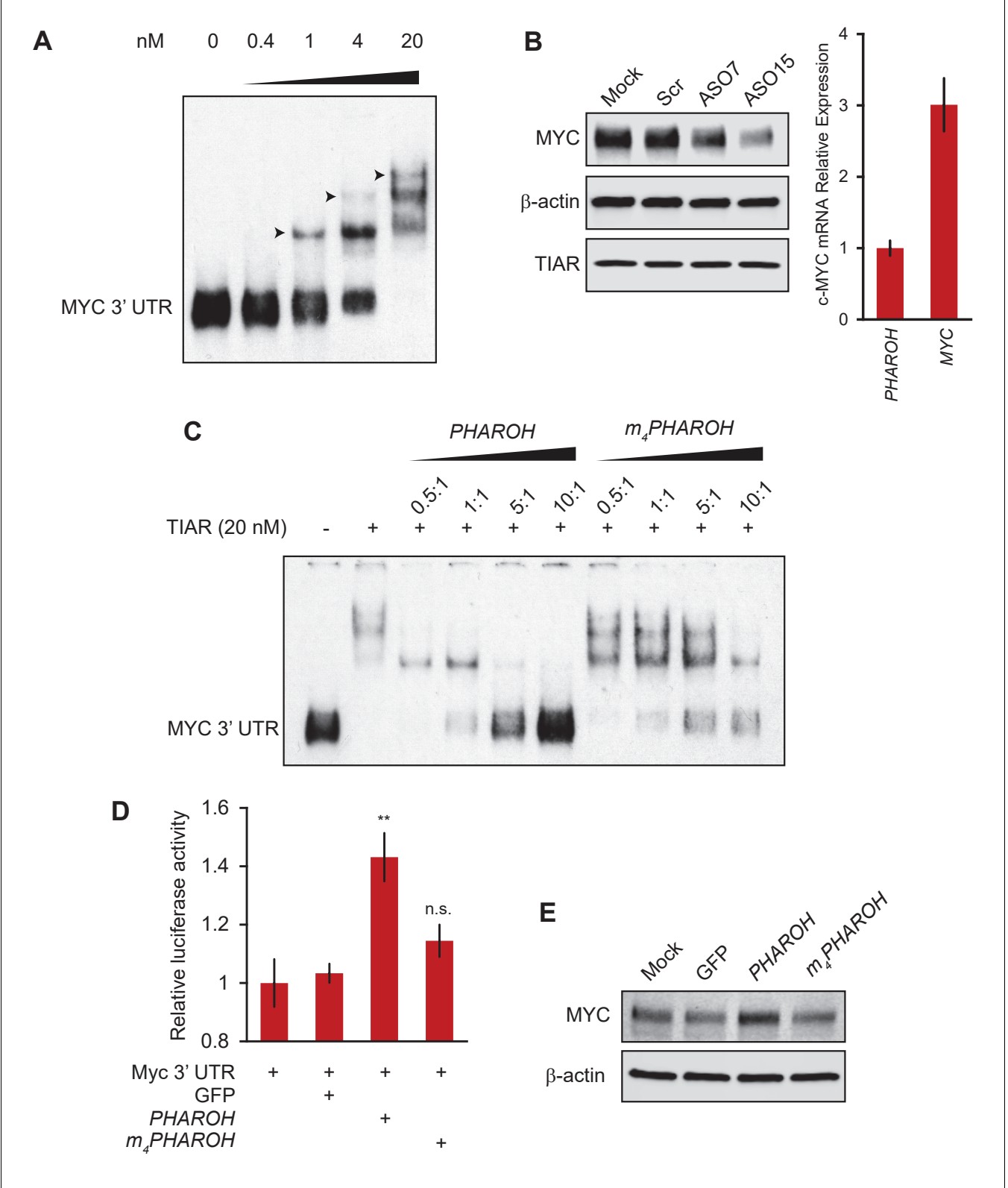

**Figure 7.** Loss of *PHAROH* releases TIAR, which inhibits Myc translation. (**A**) RNA electromobility shift assay of the 53-nt Myc 3′ UTR fragment shows that TIAR has three potential binding sites, but prefers a single binding event (note arrows). (**B**) Knockdown of *PHAROH* reduces MYC protein levels, but not TIAR levels, even though MYC is expressed threefold higher than *PHAROH*. (**C**) Wildtype *PHAROH* hairpin is able to compete out the MYC-TIAR interaction, but the mutated hairpin is not as effective in competing with the Myc-TIAR interaction. (**D**) Luciferase activity is increased with the

*Figure 7 continued on next page*

*Figure 7 continued*

addition of *PHAROH* but not with *m₄PHAROH* (**p<0.01; Student's t-test). (E) Overexpression of *PHAROH* increases MYC protein expression, but overexpression of *m₄PHAROH* does not change MYC levels appreciably.

The online version of this article includes the following figure supplement(s) for figure 7:

**Figure supplement 1.** Loss ofPHAROHreleases TIAR, which inhibits Myc translation, related to *Figure 7*.

5 µg/ml. Genomic DNA was isolated using DNeasy Blood and Tissue (Qiagen). All cell lines were tested for mycoplasma regularly.

## Cellular fractionation

Cellular fractionation was performed according to https://link.springer.com/protocol/10.1007%2F978-1-4939-4035-6_1. In brief, cells were collected and resuspended in NP-40 lysis buffer. The cell suspension is overlaid on top of a sucrose buffer and centrifuged at 3500 × g for 10 min to pellet the nuclei. The supernatant (cytoplasm) is collected and the nuclei are resuspended in glycerol buffer and urea buffer is added to separate the nucleoplasm and chromatin. The cells are centrifuged at 14,000 × g for 2 min and the supernatant (nucleoplasm) is collected, while the chromatin-RNA is pelleted.

## DEN administration

Mice were injected intraperitoneally with DEN at 14 days of age as described (*García-Irigoyen et al., 2015*). DEN-treated mice, and the corresponding controls injected with saline, were sacrificed at 5, 8, and 11 months post injection.

## Partial hepatectomy

Two-thirds PH and control sham operations (SH) were performed as reported (*Berasain et al., 2005*). Two SH and four PH mice were sacrificed at 3, 6, 24, 48, and 72 hr after surgery. Animal experimental protocols were approved (CEEA 062-16) and performed according to the guidelines of the Ethics Committee for Animal Testing of the University of Navarra.

## Human samples

Samples from patients included in the study were provided by the Biobank of the University of Navarra (CEI 47/2015) and were processed following standard operating procedures approved by the Ethical and Scientific Committees. Liver samples from healthy patients were collected from individuals with normal or minimal changes in the liver at surgery of digestive tumors or from percutaneous liver biopsy performed because of mild alterations of liver function. Samples for cirrhotic liver and HCC were obtained from patients undergoing PH and/or liver transplantation.

The biobank obtained an informed consent and consent to publish from each patient, and codified samples were provided to the researchers. The study protocol conformed to the ethical guidelines of the 1975 Declaration of Helsinki. Samples were processed following standard operating procedures approved by the Ethical and Scientific Committees. Liver samples from healthy patients were collected from individuals with normal or minimal changes in the liver at surgery of digestive tumors or from percutaneous liver biopsy performed because of mild alterations of liver function. Samples for cirrhotic liver and HCC were obtained from patients undergoing PH and/or liver transplantation.

## Immunoblotting

To determine protein levels in our system, we used 10% SDS-PAGE gels. Gels were loaded with 1 µg protein per well (Bradford assay). The following antibodies were used: β-actin (1:15,000; Sigma), c-Myc (1:1000; CST), and TIAR (1:1000; Cell Signaling). IRDye-800CW was used as a fluor for secondary anti-rabbit antibodies, and IRDye-680RD was used for mouse secondary antibodies. Blots were scanned using the Li-Cor Odyssey Classic.

## Immunoprecipitation

For TIAR immunoprecipitation, one 10 cm plate of Hepa1c1c7 cells at 80% confluence was lysed in 1 ml Pierce IP Lysis Buffer (supplemented with 100 U/ml SUPERase-IN and 1X Roche protease inhibitor cocktail) and incubated on ice for 10 min. Lysates centrifuged at 13,000 × g for 10 min. 3 μg of TIAR antibody or rabbit IgG were incubated with the lysate at 4°C for 1 hr. 16 μl of Protein A magnetic beads were washed and added to the lysate and incubated for an additional 30 min at 4°C. 50% of beads were resuspended in Laemmli buffer for western blotting and RNA was isolated from the remaining beads using TRIzol.

## Immunofluorescence staining

#1.5 round glass coverslips were prepared by acid-cleaning prior to seeding cells. Staining was performed as published previously (Spector, D.L. and H.C. Smith. 1986. Exp. Cell Res. 163, 87–94). In brief, cells were fixed in 2% PFA for 15 min, washed with PBS, and permeabilized in 0.2% Triton X-100 plus 1% normal goat serum (NGS). Cells were washed again in PBS + 1% NGS and incubated with TIAR antibody (1:2000; CST) for 1 hr at room temperature in a humidified chamber. Cells were washed again PBS + 1% NGS and incubated with Goat anti-Rabbit IgG (H + L) Cross-Adsorbed Secondary Antibody, Alexa Fluor 488 (1:1000; Thermo Fisher) secondary antibody for 1 hr at room temperature. Cover slips were washed with PBS before mounting with ProLong Diamond antifade (Thermo Fisher).

## Cell viability assays

Cells were seeded at a density of 10,000 cells/well (100 μl per well) into 24-well plates and treated with 2.5 μM of either a *PHAROH*-specific ASO or scASO. Cells were grown for 96 hr at 37°C. 20 μl of solution (CellTiter 96 AQueous One Solution Reagent, Promega) was added to the wells and incubated for 4 hr at 37°C. Measurements of absorbance at 490 nm were performed using a SpectraMax i3 Multi-Mode Detection Platform (Molecular Devices). Background absorbance at 690 nm was subtracted. Cells were also trypsinized, pelleted, and manually counting using a hemocytometer.

## RNA antisense pulldown and mass spectrometry

### RNA antisense pulldown

Cells were lysed on a 10 cm plate in 1 ml IP lysis buffer (IPLB, 25 mM Tris-HCl pH 7.4, 150 mM NaCl, 1% NP-40, 1 mM EDTA, 5% glycerol, supplemented with 100 U/ml SUPERase-IN and 1X Roche protease inhibitor cocktail) for 10 min, and lysate was centrifuged at 13,000 × g for 10 min. Cell lysate was adjusted to 0.3 mg/ml (Bradford assay). 100 pmol of biotinylated oligo was added to 500 μl of lysate and incubated at room temperature for 1 hr with rotation. 100 μl streptavidin Dynabeads were washed in IPLB, added to the lysate, and incubated for 30 min at room temperature with rotation. Beads were washed three times with 1 ml lysis buffer. For determining temperature for optimal elution, beads were then resuspended in 240 μl of 100 mM TEAB and aliquoted into eight PCR tubes. Temperature was set on a veriflex PCR block and incubated for 10 min. Beads were captured and TRIzol was added to the eluate and beads. Once optimal temperature is established, the beads were resuspended in 90 μl of 100 mM TEAB and incubated at 40°C for 10 min. TRIzol was added to 30 μl of the eluate, another 30 μl was kept for western blots, and the last 30 μl aliquot was sent directly for mass spectrometry.

### Tryptic digestion and iTRAQ labeling

Eluted samples were reduced and alkylated with 5 mM DTT and 10 mM iodoacetamide for 30 min at 55°C, then digested overnight at 37°C with 1 μg Lys-C (Promega, VA1170) and dried in vacuo. Peptides were then reconstituted in 50 μl of 0.5 M TEAB/70% ethanol and labeled with 4-plex iTRAQ reagent for 1 hr at room temperature essentially as previously described (*Ross et al., 2004*). Labeled samples were then acidified to <pH 4 using formic acid, combined and concentrated in vacuo until ~10 μl remained.

### Two-dimensional fractionation

Peptides were fractionated using a Pierce High pH Reversed-Phase Peptide Fractionation Kit (Thermo Scientific, 84868) according to the manufacturer's instructions with slight modifications.

Briefly, peptides were reconstituted in 150 µl of 0.1% TFA, loaded onto the spin column, and centrifuged at 3000 × g for 2 min. Column was washed with water, and then peptides were eluted with the following percentages of acetonitrile (ACN) in 0.1% triethylamine (TEA): 5, 7.5, 10, 12.5, 15, 20, 30, and 50%. Each of the eight fractions was then separately injected into the mass spectrometer using capillary reverse-phase LC at low pH.

## Mass spectrometry

An Orbitrap Fusion Lumos mass spectrometer (Thermo Scientific) equipped with a nano-ion spray source was coupled to an EASY-nLC 1200 system (Thermo Scientific). The LC system was configured with a self-pack PicoFrit 75 µm analytical column with an 8 µm emitter (New Objective, Woburn, MA) packed to 25 cm with ReproSil-Pur C18-AQ, 1.9 µM material (Dr. Maish GmbH). Mobile phase A consisted of 2% ACN; 0.1% formic acid, and mobile phase B consisted of 90% ACN; 0.1% formic acid. Peptides were then separated using the following steps: at a flow rate of 200 nl/min: 2% B to 6% B over 1 min, 6% B to 30% B over 84 min, 30% B to 60% B over 9 min, 60% B to 90% B over 1 min, held at 90% B for 5 min, 90% B to 50% B over 1 min, and then flow rate was increased to 500 µl/min as 50% B was held for 9 min. Eluted peptides were directly electrosprayed into the MS with the application of a distal 2.3 kV spray voltage and a capillary temperature of 300°C. Full-scan mass spectra (Res = 60,000; 400–1600 m/z) were followed by MS/MS using the 'Top Speed' method for selection. High-energy collisional dissociation (HCD) was used with the normalized collision energy set to 35 for fragmentation, the isolation width set to 1.2, and a duration of 15 s was set for the dynamic exclusion with an exclusion mass width of 10 ppm. We used monoisotopic precursor selection for charge states 2+ and greater, and all data were acquired in profile mode.

## Database searching

Peaklist files were generated by Proteome Discoverer version 2.2.0.388 (Thermo Scientific). Protein identification was carried out using both Sequest HT (*Eng et al., 1994*) and Mascot 2.5 (*Perkins et al., 1999*) against the UniProt mouse reference proteome (57,220 sequences; 26,386,881 residues). Carbamidomethylation of cysteine, iTRAQ4plex (K), and iTRAQ4plex (N-term) was set as fixed modifications, methionine oxidation and deamidation (NQ) were set as variable modifications. Lys-C was used as a cleavage enzyme with one missed cleavage allowed. Mass tolerance was set at 20 ppm for intact peptide mass and 0.3 Da for fragment ions. Search results were rescored to give a final 1% FDR using a randomized version of the same Uniprot mouse database, with two peptide sequence matches (PSMs) required. iTRAQ ratio calculations were performed using Unique and Razor peptide categories in Proteome Discoverer.

## RNA EMSA

DNA template used for in vitro synthesis of RNA probes were from annealed oligos. A T7 RNA polymerase promoter sequence was added to allow for in vitro transcription using the MEGAscript T7 transcription kit (Thermo Fisher). RNA was end labeled at the 3′ end with biotin using the Pierce RNA 3′ End Biotinylation Kit (Thermo Fisher). RNA quantity was assayed by running an RNA 6000 Nano chip on a 2100 Bioanalyzer. 6% acrylamide gels (39:1 acrylamide:bis) (Bio-Rad) containing 0.5 × TBE were used for all EMSA experiments. Recombinant human TIAR (Proteintech) was added at indicated concentrations to the probe (~2 fmol) in 20 µl binding buffer, consisting of 10 mM HEPES (pH 7.3), 20 mM KCL, 1 mM $Mg_2Cl_2$, 1 mM DTT, 30 ng/µl BSA, 0.01% NP-40, and 5% glycerol. After incubation at room temperature for 30 min, 10 µl of the samples were loaded and run for 1 hr at 100 V. The nucleic acids were then transferred onto a positively charged nylon membrane (Amersham Hybond-N+) in 0.5 × TBE for 30 min at 40 mAh. Membranes were crosslinked using a 254 nM bulb at 120 mJ/cm² in a Stratalinker 1800. Detection of the biotinylated probe was done using the Chemiluminescent Nucleic Acid Detection Module Kit (Thermo Fisher 89880).

## 3′ UTR luciferase assay

The full-length 3′ UTR of c-Myc was cloned into the pmirGLO Dual-Luciferase miRNA target expression vector (Promega). Luciferase activity was assayed in transfected cells using the Dual-Luciferase Reporter Assay (Promega). To evaluate the interaction between *PHAROH*, 3′ UTR of c-Myc, and TIAR, cells were transfected with the respective constructs using Lipofectamine 3000. 24 hr later,

firefly and Renilla luciferase activity was measured, and Renilla activity was used to normalize firefly activity.

## Single-molecule RNA FISH

#1.5 round glass coverslips were prepared by acid-cleaning and layered with gelatin for 20 min, prior to seeding MEF feeder cells and ESCs. Cells were fixed for 30 min in freshly prepared 4% PFA (Electron Microscopy Sciences), diluted in D-PBS without $CaCl_2$ and $MgCl_2$ (Gibco, Life Technologies), and passed through a 0.45 µm sterile filter. Fixed cells were dehydrated and rehydrated through an ethanol gradient (50–75% – 100–75% – 50% – PBS) prior to permeabilization for 5 min in 0.5% Triton X-100. Protease QS treatment was performed at a 1:8000 dilution. QuantiGene ViewRNA (Affymetrix) probe hybridizations were performed at 40°C in a gravity convection incubator (Precision Scientific), and incubation time of the pre-amplifier was extended to 2 hr. Nuclei were counter-stained with DAPI and coverslips mounted in Prolong Gold anti-face medium (http://www.spectorlab.labsites.cshl.edu/protocols).

Coverslips were imaged on a DeltaVision Core system (Applied Precision), based on an inverted IX-71 microscope stand (Olympus) equipped with a 60× U-PlanApo 1.40 NA oil immersion lens (Olympus). Images were captured at 1 × 1 binning using a CoolSNAP HQ CCD camera (Photometric) as z-stacks with a 0.2 µm spacing. Stage, shutter, and exposure were controlled through SoftWorx (Applied Precision). Image deconvolution was performed in SoftWorx.

A spinning-disc confocal system (UltraVIEW Vox; PerkinElmer) using a scanning unit (CSU-X1; Yokogawa Corporation of America) and a charge-coupled device camera (ORCA-R2; Hamamatsu Photonics) were fitted to an inverted microscope (Nikon) equipped with a motorized piezoelectric stage (Applied Scientific Instrumentation). Image acquisition was performed using Volocity versions 5 and 6 (PerkinElmer). Routine imaging was performed using Plan Apochromat 60 or 100× oil immersion objectives, NA 1.4.

## RNA sequencing and analysis

Total RNA was isolated either directly from cryosections of the tumor tissue or from organotypic epithelial cultures using TRIzol according to the manufacturer's instructions. RNA quality was assayed by running an RNA 6000 Nano chip on a 2100 Bioanalyzer. For high-throughput sequencing, RNA samples were required to have an RNA integrity number (RIN) 9 or above. TruSeq (Illumina) libraries for poly(A)+RNA seq were prepared from 0.5 to 1 mg RNA per sample. To ensure efficient cluster generation, an additional gel purification step of the libraries was applied. The libraries were multiplexed (12 libraries per lane) and sequenced single-end 75 bp on the NextSeq500 platform (Illumina), resulting in an average 40 million reads per library. Analysis was performed in GalaxyProject. In brief, reads were first checked for quality using FastQC (http://www.bioinformatics.babraham.ac.uk/projects/fastqc/), and a minimum Phred score of 30 was required. Reads were then mapped to the mouse mm10 genome using STAR (*Dobin et al., 2013*), and counts were generating using htseq-counts with the appropriate GENCODE M20 annotation. Deseq2 was then used to generate the list of differentially expressed genes (*Love et al., 2014*). Motif analysis was performed using HOMER (*Heinz et al., 2010*).

Coding analysis cDNA sequences of *PHAROH* and *GAPDH* were inputted into CPAT (http://lilab.research.bcm.edu/cpat/) or CPC (http://cpc.cbi.pku.edu.cn/programs/run_cpc.jsp) for analysis (*Kong et al., 2007*; *Wang et al., 2013*). PhyloCSF analysis was performed using the UCSC Genome Browser track hub (https://data.broadinstitute.org/compbio1/PhyloCSFtracks/trackHub/hub.DOC.html) (*Lin et al., 2011*).

Plasmid construction eSpCas9(1.1) was purchased from Addgene (#71814). eSpCas9-2A-GFP was constructed by subcloning 2A-GFP from pSpCas9(BB)−2A-GFP (PX458) (Addgene #48138) into eSpCas9 using EcoRI sites. To construct eSpCas9-2A-mCherry, 2A-mCherry was amplified from mCherry-Pol II (*Zhao et al., 2011*), and an internal BbsI site was silently mutated. The PCR product was then cloned into eSpCas9 using EcoRI sites. The *PHAROH* construct was amplified using Hepa1-6 cDNA as a template and cloned into pCMV6 using BamHI and FseI. Mutant *PHAROH* was constructed by amplifying tiled oligos and cloned into pCMV6 using BamHI and FseI.

## CRISPR/Cas9 genetic knockout

To generate a genetic knockout of *PHAROH*, two sgRNAs targeting the promoter region were combined, creating a deletion including the TSS. Guide design was performed on Benchling (https://benchling.com) taking into account both off-target scores and on-target scores. The sgRNA targeting the gene body of *PHAROH* was cloned into a pSpCas9(BB)−2A-GFP vector (PX458, Addgene plasmid #48138), and the sgRNA targeting the upstream promoter region was cloned into a pSpCas9(BB)- 2A-mCherry vector. Hepa1-6 were transfected with both plasmids using the 4D-Nucleofector System (Lonza) using the EH-100 program in SF buffer. To select for cells expressing both gRNAs, GFP and mCherry double-positive cells were sorted 48 hr post transfection as single-cell deposition into 96-well plates using a FACS Aria (SORP) Cell Sorter (BD). Each single-cell clone was propagated and analyzed by genomic PCR and qRT-PCR to select for homozygous knockout clones. Cells transfected with a sgRNA targeting Renilla luciferase were used as a negative control.

## Cell cycle analysis

Hoechst 33342 (Sigma) was added to cells at a final concentration of 5 µg/ml and incubated at 37°C for 1 hr. Cells were trypsinized and collected into a flow cytometry compatible tube. Profiles were analyzed using a FACS Aria (SORP) Cell Sorter (BD), gated according to DNA content and cell cycle phase, and sorted into Eppendorf tubes for subsequent RNA extraction and qRT-PCR analysis.

## Nucleofection

For transfection of ASOs using nucleofection technology (Lonza), ESCs were harvested following soaking off of feeder cells for 1 hr, washed in D-PBS (Gibco, Life Technologies), and passed through a 70 µm nylon cell strainer (Corning). Cell count and viability was determined by trypan blue staining on a Countess automated cell counter (Life Technologies). For each reaction, $1 \times 10^6$ viable cells were resuspended in SF Cell Line Solution (Lonza), mixed with 2 µM control or 2 µM target-specific ASO, and transferred to nucleocuvettes for nucleofection on a 4D-Nucleofector System (Lonza) using program code 'EH-100'. For plasmid nucleofections, 10 µg of plasmid was used and nucleofected using program code 'EH-100'. Cells were subsequently transferred onto gelatinized cell culture plates containing pre-warmed and supplemented growth medium. Growth medium was changed once after 16 hr.

## Colony formation assay

200 Hepa1-6 cells were seeded in a 6-well plate. ASOs were added at the time of seeding at the indicated concentrations. Two weeks later, cells were fixed, stained with Giemsa, counted, and photographed.

## 2′-O-Methoxyethyl (MOE) ASOs and knockdown analysis

Synthesis and purification of all 2′-MOE modified oligonucleotides was performed as previously described (*Meng et al., 2015*) by Ionis Pharmaceuticals. These ASOs are 20-mer oligonucleotides containing a phosphorothioate backbone, 2′-MOE modifications on the first and last five nucleotides and a stretch of 10 DNA bases in the center. Constrained ethyl oligos are 16-mer oligonucleotides that contain modifications on the first and last three nucleotides and a stretch of 10 DNA bases in the center.

## qRT-PCR

To assess knockdown efficiency, TRIzol-extracted RNA was treated with RNAse-free DNAseI (Life Technologies) and subsequently reverse-transcribed into cDNA using TaqMan Reverse Transcription reagents and random hexamer oligonucleotides (Life Technologies). Real-time PCR reactions were prepared using Power SYBR Green Master Mix (Life Technologies) and performed on an ABI StepOnePlus Real-Time PCR system (Life Technologies) for 40 cycles of denaturation at 95°C for 15 s followed by annealing and extension at 60°C for 60 s. Primers were designed to anneal within an exon to detect both primary and processed transcripts. Primer specificity was monitored by melting curve analysis. For each sample, relative abundance was normalized to the housekeeping gene *PPIB* mRNA levels.

## Acknowledgements

We thank members of the Spector lab for critical discussions and advice throughout the course of this study. We would also like to thank the CSHL Cancer Center Shared Resources (Microscopy, Mass Spectrometry, Flow Cytometry, and Next-Gen Sequencing) for services and technical expertise (NCI 2P3OCA45508). This research was supported by NCI 5P01CA013106-Project 3 and NIGMS 5R35GM131833 (DLS) and NCI 5F31CA220997-02 (ATY).

## Additional information

### Competing interests

Frank Rigo: Employee of Ionis Pharmaceuticals. David L Spector: DLS is a consultant to and receives research reagents from Ionis Pharmaceuticals. The other authors declare that no competing interests exist.

### Funding

| Funder | Grant reference number | Author |
| --- | --- | --- |
| National Cancer Institute | 5PO1CA013106-Project 3 | David L Spector |
| National Cancer Institute | 5F31CA220997 | Allen T Yu |
| National Cancer Institute | 2P3OCA45508 | David L Spector |
| National Institute of General Medical Sciences | 5R35GM131833 | David L Spector |

The funders had no role in study design, data collection and interpretation, or the decision to submit the work for publication.

### Author contributions

Allen T Yu, Formal analysis, Funding acquisition, Investigation, Methodology, Writing - original draft; Carmen Berasain, Formal analysis, Investigation, Methodology, Writing - review and editing; Sonam Bhatia, Formal analysis, Investigation, Writing - review and editing; Keith Rivera, Darryl J Pappin, Formal analysis, Methodology, Writing - review and editing; Bodu Liu, Methodology, Writing - review and editing; Frank Rigo, Writing - review and editing, Provided antisense reagents; David L Spector, Conceptualization, Supervision, Funding acquisition, Project administration, Writing - review and editing

### Author ORCIDs

Allen T Yu https://orcid.org/0000-0002-1842-857X
Sonam Bhatia http://orcid.org/0000-0002-0124-2621
David L Spector https://orcid.org/0000-0003-3614-4965

### Ethics

Human subjects: The Human Research Review Committee of the University of Navarra (CEI 47/2015) approved the study and human samples were provided by the Biobank of the University of Navarra. The biobank obtained an informed consent and consent to publish from each patient and codified samples were provided to the researchers. The study protocol conformed to the ethical guidelines of the 1975 Declaration of Helsinki. Samples were processed following standard operating procedures approved by the Ethical and Scientific Committees. Liver samples from healthy patients were collected from individuals with normal or minimal changes in the liver at surgery of digestive tumors or from percutaneous liver biopsy performed because of mild alterations of liver function. Samples for cirrhotic liver and HCC were obtained from patients undergoing partial hepatectomy and/or liver transplantation.

Animal experimentation: Animal experimental protocols were approved (CEEA 062-16) and performed according to the guidelines of the Ethics Committee for Animal Testing of the University of Navarra.

Decision letter and Author response
Decision letter https://doi.org/10.7554/eLife.68263.sa1
Author response https://doi.org/10.7554/eLife.68263.sa2

## Additional files

### Supplementary files
• Transparent reporting form

### Data availability
RNA-seq data has been uploaded to GEO: GSE167316.

The following dataset was generated:

| Author(s) | Year | Dataset title | Dataset URL | Database and Identifier |
|---|---|---|---|---|
| Yu AT, Spector DL | 2021 | PHAROH lncRNA regulates c-Myc translation in hepatocellular carcinoma via sequestering TIAR | https://www.ncbi.nlm.nih.gov/geo/query/acc.cgi?acc=GSE167316 | NCBI Gene Expression Omnibus, GSE167316 |

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

# Appendix 1

**Appendix 1—key resources table**

| Reagent type (species) or resource | Designation | Source or reference | Identifiers | Additional information |
|---|---|---|---|---|
| Strain, strain background (C57BL/6J) (*Mus musculus*) Female | C57BL/6J | The Jackson Laboratory | Stock No: 000664 RRID:IMSR_JAX:000664 | |
| Gene (*Homo sapiens*) | Tial1 (NM_009383) Mouse Tagged ORF Clone | Origene | Cat# MG226372 | |
| Gene (*Mus musculus*) | Myc | GenBank | NC_000081.7 | |
| Recombinant Protein (*Homo sapiens*) | Recombinant Human TIAL1 Protein | Novus Biologicals | Cat# NBP2-51914-0.1mg | |
| Cell line (*Mus musculus*) | AB2.2 (ESCs) | ***Bergmann et al., 2015*** | | Cell line maintained in D. L. Spector Lab |
| Cell line (*Mus musculus*) | NPC | ***Bergmann et al., 2015*** | | Cell line maintained in D. L. Spector Lab |
| Cell line (*Mus musculus*) | Hepa1-6 | ATCC | Cat# CRL-1830 | Cell line maintained in D. L. Spector Lab |
| Cell line (*Mus musculus*) | Hepa1c1c7 | ATCC | Cat# CRL-2026 | Cell line maintained in D. L. Spector Lab |
| Cell line (*Mus musculus*) | AML12 | ATCC | Cat# CRL-2254 | Cell line maintained in D. L. Spector Lab |
| Cell line (*Mus musculus*) | MEF | MTI-Global Stem | Cat# GSC-6601G | Irradiated feeder MEFs |
| Cell line (*Homo sapiens*) | SNU-182 | ATCC | Cat# CRL-2235 | Cell line maintained in D. L. Spector Lab |
| Cell line (*Homo sapiens*) | Huh1 | N/A | | Generous gift from Scott Lowe (MSKCC) |
| Cell line (*Homo sapiens*) | Huh7 | N/A | | Generous gift from Scott Lowe (MSKCC) |
| Cell line (*Homo sapiens*) | JHH2 | N/A | | RNA gifted from Scott Lowe (MSKCC) |
| Cell line (*Homo sapiens*) | SNU-387 | ATCC | Cat# CRL-2237 | Generous gift from Scott Lowe (MSKCC) |
| Cell line (*Homo sapiens*) | Hep3B | ATCC | Cat# HB-8064 | Generous gift from Scott Lowe (MSKCC) |
| Cell line (*Homo sapiens*) | Alex | ATCC | Cat# CRL-8024 | RNA gifted from Scott Lowe (MSKCC) |
| Cell line (*Homo sapiens*) | HepG2 | ATCC | Cat# HB-8065 | Generous gift from Scott Lowe (MSKCC) |
| Cell line (*Homo sapiens*) | Li7 | N/A | | RNA gifted from Scott Lowe (MSKCC) |
| Cell line (*Homo sapiens*) | THLE-2 | ATCC | Cat# CRL-2706 | Cell line maintained in D. L. Spector Lab |
| Antibody | c-Myc (rabbit monoclonal) | Cell Signaling | Cat# 5605 RRID:AB_1903938 | (IB: 1:1000) |

*Continued on next page*

*Appendix 1—key resources table continued*

| Reagent type (species) or resource | Designation | Source or reference | Identifiers | Additional information |
|---|---|---|---|---|
| Antibody | TIAR (rabbit monoclonal) | Cell Signaling | Cat# 8509 RRID:AB_10839263 | (IB: 1:1000) (IF: 1:2000) (IP: 1:100) |
| Antibody | β-Actin (mouse monoclonal) | Cell Signaling | Cat# 3700 RRID:AB_2242334 | (IB: 1:10,000) |
| Antibody | IRDye 800CW (Goat anti-Rabbit IgG) | LI-COR Biosciences | Cat# 925-32211 RRID:AB_2651127 | (IB: 1:10,000) |
| Antibody | IRDye 680RD (Goat anti-Mouse IgG) | LI-COR Biosciences | Cat# 925-68070 RRID:AB_2651128 | (IB: 1:10,000) |
| Antibody | Goat anti-Rabbit IgG (H + L) Cross-Adsorbed Secondary Antibody Alexa Fluor 488 | Thermo Fisher | Cat# A-11008 RRID:AB_143165 | (IF: 1:1000) |
| Antibody | Rabbit IgG Isotype Control | Thermo Fisher | Cat# 10500C RRID:AB_2532981 | |
| Recombinant DNA reagent | eSpCas9-1.1 | Addgene | RRID:Addgene_71814 | Backbone for constructing GFP and mCherry variants |
| Recombinant DNA reagent | eSpCas9-1.1-GFP (plasmid) | This study | N/A | |
| Recombinant DNA reagent | eSpCas9-1.1-mCherry (plasmid) | This study | N/A | |
| Recombinant DNA reagent | pmirGLO | Promega | Cat# E1330 | Dual-Luciferase miRNA Target Expression Vector |
| Recombinant DNA reagent | pCMV6-A-Puro | Origene | Cat# PS100025 | pCMV6 backbone |
| Transfected construct (*Mus musculus*) | sgPHAROH_F-eSpCas9-1.1-GFP (plasmid) | This study | N/A | Upstream PHAROH sgRNA |
| Transfected construct (*Mus musculus*) | sgPHAROH_R-eSpCas9-1.1-mCherry (plasmid) | This study | N/A | Downstream PHAROH sgRNA |
| Transfected construct (*Mus musculus*) | sgRenilla-eSpCas9-1.1-GFP (plasmid) | *Chang et al., 2020* | N/A | Negative control sgRNA |
| Transfected construct (*Mus musculus*) | pmirGLO-MYC (plasmid) | This study | N/A | Construct for luciferase assay readout |
| Transfected construct (*Mus musculus*) | pCMV6-pharoh (plasmid) | This study | N/A | Construct for rescue and luciferase assay readout |
| Transfected construct (*Mus musculus*) | pCMV6-m4pharoh (plasmid) | This study | N/A | Construct for luciferase assay readout |
| Transfected construct (*Mus musculus*) | pCMV6-GFP (plasmid) | *Chang et al., 2020* | N/A | Construct for luciferase assay readout |

*Continued on next page*

*Appendix 1—key resources table continued*

| Reagent type (species) or resource | Designation | Source or reference | Identifiers | Additional information |
|---|---|---|---|---|
| Sequence-based reagent | ASO 7 | This study | *PHAROH* Gapmer ASO | CGTGTCATCTTCTTGGCCCC |
| Sequence-based reagent | ASO 15 | This study | *PHAROH* Gapmer ASO | TCGTGTCATCTTCTTGGCCC |
| Sequence-based reagent | ASO 14 | This study | *PHAROH* cEt ASO | GTTACAGGACGCATGT |
| Sequence-based reagent | ASO 18 | This study | *PHAROH* cEt ASO | CACATAGTTATTCCCG |
| Sequence-based reagent | Forward | This study | *PHAROH* genomic PCR | TGCTTAGCACGT CCTCAGTGC |
| Sequence-based reagent | Reverse | This study | *PHAROH* genomic PCR | AGTTCCCCAGC AACCCTGTT |
| Sequence-based reagent | Upstream | This study | *PHAROH* sgRNA | GCAGGTAGTGT GGTAACTCC |
| Sequence-based reagent | Downstream | This study | *PHAROH* sgRNA | CGGGTCCTCCC AGCGCACAC |
| Sequence-based reagent | Exon 4 Fwd | This study | *PHAROH* qRT-PCR | GGGGCCAAGAA GATGACACG |
| Sequence-based reagent | Exon 4 Ref | This study | *PHAROH* qRT-PCR | GGACGCATGT GGAGGTCAGA |
| Sequence-based reagent | Exon A Fwd | This study | *PHAROH* qRT-PCR | TGCCTCACAA GGGACAACACTC |
| Sequence-based reagent | Exon A Rev | This study | *PHAROH* qRT-PCR | GAATTTGCTCA GGGGCTCCA |
| Sequence-based reagent | Exon B Fwd | This study | *PHAROH* qRT-PCR | GGACTTGAACT GGCACTGTTGC |
| Sequence-based reagent | Exon B Rev | This study | *PHAROH* qRT-PCR | CAGAAGGACC ATCATCACGA |
| Sequence-based reagent | Exon C Fwd | This study | *PHAROH* qRT-PCR | TGAACCCGAGC TTTGCCATT |
| Sequence-based reagent | Exon C Rev | This study | *PHAROH* qRT-PCR | CGGTGCTCTG CAGGACGTTT |
| Sequence-based reagent | Exon D Fwd | This study | *PHAROH* qRT-PCR | AGGCTGCCGC CACACTTAAA |
| Sequence-based reagent | Exon D Rev | This study | *PHAROH* qRT-PCR | TTCAGCTGCTGG CATTCTTCC |
| Sequence-based reagent | Exon E Fwd | This study | *PHAROH* qRT-PCR | GGAGAGAACAA GGGCCTTCC |
| Sequence-based reagent | Exon E Rev | This study | *PHAROH* qRT-PCR | GCCCTGCTGCA TTCTGGGTA |
| Sequence-based reagent | Exon 1 Fwd | This study | *PHAROH* qRT-PCR | GGTGTGAACCAA GTGCACGTCT |
| Sequence-based reagent | Exon 1 Rev | This study | *PHAROH* qRT-PCR | GGGATCTGACA CCGCCTTCTT |
| Sequence-based reagent | Exon 2 Fwd | This study | *PHAROH* qRT-PCR | CTTCTGAGTCTG ACGGGCTGGT |
| Sequence-based reagent | Exon 2 Rev | This study | *PHAROH* qRT-PCR | TCAGTCCTACCC AAGAAATTTAGGA |
| Sequence-based reagent | Exon 3 Fwd | This study | *PHAROH* qRT-PCR | TGTGGAAACTCA GAGAGGATGC |

*Appendix 1—key resources table continued*

| Reagent type (species) or resource | Designation | Source or reference | Identifiers | Additional information |
|---|---|---|---|---|
| Sequence-based reagent | Exon 3 Rev | This study | *PHAROH* qRT-PCR | CTCTGGTGGCTG TGCCTTCAAA |
| Sequence-based reagent | MycF | This study | Myc qRT-PCR | CAACGTCTTGG AACGTCAGA |
| Sequence-based reagent | MycR | This study | Myc qRT-PCR | TCGTCTGCTT GAATGGACAG |
| Sequence-based reagent | Outer 1 | This study | 5' RACE | TTCCTGCGTG AAAGTGTCTG |
| Sequence-based reagent | Outer 2 | This study | 5' RACE | TGACCTTCTCA GGAAGTGGAA |
| Sequence-based reagent | Inner 1 | This study | 5' RACE | CCTGAGAGGAC GAGGTGACT |
| Sequence-based reagent | Inner 2 | This study | 5' RACE | TTTGCAGGTTA GGATCAGAGC |
| Sequence-based reagent | Outer | This study | 3' RACE | CACTTCCATT CCTCCCCATA |
| Sequence-based reagent | Inner | This study | 3' RACE | GGGGACTCAGA CACTCACCA |
| Sequence-based reagent | PHAROH hairpin | This study | T7 Transcription Primer | TAATACGAC TCACTATA gagaggatgccactgttttg aactattttgaaggcacag ccaccagagctttaggg acagggtattttatc |
| Sequence-based reagent | Myc 3' UTR | This study | T7 Transcription Primer | TAATACGACTCACTATAG cttcccatcttttttcttttccc ttttaacagatttg tatttaattgttttt |
| Sequence-based reagent | m1 | This study | T7 Transcription Primer | TAATACGACTCACTATA gagaggatgccactgtCt Cgaactattttgaaggca cagccaccagagctta gggacagggtattttatc |
| Sequence-based reagent | m2 | This study | T7 Transcription Primer | TAATACGACTCACTATA gagaggatgccactgtCtC gaactaCtCtgaaggcac agccaccagagctttaggg acagggtattttatc |
| Sequence-based reagent | m3 | This study | T7 Transcription Primer | TAATACGACTCACTATA gagaggatgccactgtCtC gaactaCtCtgaaggcac agccaccagagcCtta gggacagggtattttatc |
| Sequence-based reagent | m4 | This study | T7 Transcription Primer | TAATACGACTCACTATA gagaggatgccactgtCtC gaactaCtCtgaaggcaca gccaccagagcCttaggg acagggtatCCtatc |
| Sequence-based reagent | PHAROH 1 | This study | Biotin antisense pulldown oligo | AGAAATTTAGGAG CCACGCT |
| Sequence-based reagent | PHAROH 2 | This study | Biotin antisense pulldown oligo | GCTGTGCCTTC AAAATAGTT |
| Sequence-based reagent | PHAROH 3 | This study | Biotin antisense pulldown oligo | GCCCCAAGAAA CTCAAGAAT |

*Appendix 1—key resources table continued*

| Reagent type (species) or resource | Designation | Source or reference | Identifiers | Additional information |
|---|---|---|---|---|
| Sequence-based reagent | PHAROH 4 | This study | Biotin antisense pulldown oligo | TTAATTTTCT CCTTTATGCA |
| Sequence-based reagent | PHAROH 5 | This study | Biotin antisense pulldown oligo | ACAACGTGTGG ATGTGTGTT |
| Sequence-based reagent | PPIB 1 | This study | Biotin antisense pulldown oligo | CCTACAGATT CATCTCCAAT |
| Sequence-based reagent | PPIB 2 | This study | Biotin antisense pulldown oligo | GTTATGAAGAA CTGTGAGCC |
| Commercial assay or kit | DNase I, Amplification Grade | Life Technologies | Cat# 18068 | |
| Commercial assay or kit | TaqMan Reverse Transcription Reagents | Thermo Fisher | Cat# 4304134 | |
| Commercial assay or kit | SF Cell Line 4D-Nucleofector X Kit L | Lonza | Cat# V4XC-2024 | |
| Commercial assay or kit | View ISH Cell Assay Kit | Affymetrix | Cat# QVC0001 | |
| Commercial assay or kit | MEGAscript T7 Transcription Kit | Thermo Fisher | AM1333 | |
| Commercial assay or kit | Pierce RNA 3' End Biotinylation Kit | Thermo Fisher | Cat# 20160 | |
| Commercial assay or kit | LightShift Chemiluminescent RNA EMSA Kit | Thermo Fisher | Cat# 20158 | |
| Commercial assay or kit | Pierce BCA Protein Assay Kit | Life Technologies | Cat# 23227 | |
| Commercial assay or kit | CellTiter 96 AQueous One Solution Cell Proliferation Assay | Promega | Cat# G3582 | |
| Commercial assay or kit | SMARTer RACE 5'/3' Kit | Takara | Cat# 634858 | |
| Commercial assay or kit | Promega Dual-Luciferase Reporter Assay System | Promega | Cat# E1960 | |
| Commercial assay or kit | DNeasy Blood and Tissue kit | Qiagen | Cat# 69504 | |
| Software, algorithm | Benchling | https://www.benchling.com/ | | Used for sgRNA design and cloning |
| Software, algorithm | CPAT | doi: 10.1093/nar/gkt006 | | |
| Software, algorithm | CPC | doi: 10.1093/nar/gkm391 | | |
| Software, algorithm | PhyloCSF | doi: 10.1093/bioinformatics/btr209 | | |

*Continued on next page*

*Appendix 1—key resources table continued*

| Reagent type (species) or resource | Designation | Source or reference | Identifiers | Additional information |
|---|---|---|---|---|
| Software, algorithm | FastQC | https://www.bioinformatics.babraham.ac.uk/projects/fastqc/ | RRID:SCR_014583 | |
| Software, algorithm | STAR | doi: 10.1002/0471250953.bi1114s51 | RRID:SCR_004463 | |
| Software, algorithm | DESeq2 | doi: 10.1186/s13059-014-0550-8 | RRID:SCR_015687 | |
| Software, algorithm | Volocity 3D Image Analysis Software | Perkin Elmer | RRID:SCR_002668 | |
| Software, algorithm | SoftWoRx | SoftWoRx Software | RRID:SCR_019157 | |
| Software, algorithm | Sequest HT | doi: 10.1016/1044-0305(94)80016-2 | | |
| Software, algorithm | Mascot 2.5 | doi: 10.1002/(SICI)1522-2683(19991201)20:18<3551::AID-ELPS3551>3.0.CO;2–2 | RRID:SCR_014322 | |
| Software, algorithm | HOMER Suite | doi: 10.1016/j.molcel.2010.05.004 | RRID:SCR_010881 | |
| Software, algorithm | Image Studio Software | LI-COR | RRID:SCR_015795 | |
| Software, algorithm | RNAfold | doi: 10.1093/nar/gkg599 | RRID:SCR_008550 | |
| Software, algorithm | mFold | doi: 10.1093/nar/gkg595 | RRID:SCR_008543 | |
| Software, algorithm | ImageJ | NIH, Bethesda, MD | RRID:SCR_003070 | |
| Chemical compound, drug | Hoechst dye | Thermo Fisher | Cat# 62249 | 1 µg/ml |
| Chemical compound, drug | DAPI | Life Technologies | Cat# D1306 | 1 µg/ml |
| Chemical compound, drug | α-Amanitin | Sigma-Aldrich | Cat# A2263 | 5 µg/ml |
| Chemical compound, drug | Diethylnitrosamine | Sigma-Aldrich | Cat# 73861 | 25 mg/kg |

