## [Decision Letter]

**Acceptance summary:**

This manuscript identifies a novel long non coding RNA, PHAROH, which is expressed in embryonic stem cells as well as hepatocellular carcinoma. Depletion of PHAROH reduced cell proliferation, which was rescued by exogenous expression of PHAROH. Genes that exhibited altered expression in PHAROH depleted cells harbored consensus upstream sequences, which matches the MYC-binding motifs. The protein expression of MYC was downregulated in PHAROH depleted cells without changes in its mRNA expression, suggesting that PHAROH regulates MYC translation. TIAR, a translation inhibitor was identified as a binding partner of PHAROH which indicates PHAROH may sequester TIAR to keep the expression of MYC at a high levels during proliferation.

**Decision letter after peer review:**

Congratulations, we are pleased to inform you that your article, "PHAROH lncRNA regulates c-Myc translation in hepatocellular carcinoma via sequestering TIAR", has been accepted for publication in *eLife* pending minor changes to the manuscript as suggested by reviewers that do not require rereview. Your article has been reviewed by 3 reviewers, one of whom is a member of our Board of Reviewing Editors, and the evaluation has been overseen by a Senior Editor.

*Reviewer #1:*

In this work, the authors attempt to determine how LNC RNAs may be involved in oncogenic progression, in this particular case, hepatocellular carcinoma (HCC).

They find, using several databases, that several LNC RNAs are dysregulated and choose the highest one, in 13% of the cases to investigate in detail for mechanistic correlation with the pathology. In a straightforward analysis, they determine that this particular LNC, which they term PHAROH plays a role in cell proliferation, as its knockout and knockdown leads to a decrease in cell proliferation in culture, which is restored by transfection into the cells. Comparing transcriptomics of control and knockout cells, a large number of mRNAs are affected, but they determine that c-Myc promoter elements are enriched in the genes identified, although c-Myc is not among them. RNA pull-down followed by mass spec identifies a protein, TIAR, a known translational repressor that binds to several short sequences in the LNC RNA. The authors postulate that the role of PHAROH is to efficiently compete for binding of the translational repressor with Myc so that c-Myc translation is enhanced without affecting the mRNA levels. Mutation of these binding sites results in decreased proliferation, and importantly oncogenesis. The data supports this model as far as it goes in the cultured cells but whether it plays a role in HCC is yet to be determined. The correlations with cirrhotic human liver seems more compelling in Figure 1D. The focus on PHAROH and TIAR, while revealing may represent only a small part of the total picture and c-Myc one of many possible targets that influence cell proliferation. Nonetheless, this work provides convincing evidence for the model where a LNC RNA can serve as a competitor for a translation regulator and a starting point for disease relevance.

The work is convincing within the context of the experimental design involving these particular players selected from a large number of possibilities. While it would not have been possible to investigate many of the other LNCs, proteins, targets, promoter elements etc., the manuscript needs to have the usual caveats in the text concerning the limitations on interpreting its relevance for a complex process such as cancer progression into HCC. The authors have not overextended their claims but they do suggest that PHAROH might be a therapeutic target. Hopefully they're right, but many of these approaches have not played out as expected, due to the large heterogeneity of tumor cells, herein supported by the fact that PHAROH is not overexpressed in the majority of the samples. Whether it is a small pathway in a random subset of cells, or an important one remains to be determined.

*Reviewer #2:*

In this study, Yu et al. use a combination of complementary methodologies including transcriptomics, biochemistry and imaging to identify and characterize the role of the long non-coding RNA (lncRNA), named PHAROH, in hepatocellular carcinoma. The authors show that PHAROH lncRNA is upregulated in hepatocellular carcinoma, and that by binding and sequestering the translational inhibitor TIAR -via a hairpin structure- enhances c-MYC mRNA translation. This regulation mechanism has an impact in cell proliferation and, thus, it has potential high clinical relevance. Overall, the biological insights into the molecular mechanism by this study are significant and potentially interesting to liver researchers and others in the scientific community.

The conclusions of this work are mostly well supported by data. The data is clear and well presented.

– Statistical analysis and P-values. The authors should add p-values to Figures: 2C, 2G, 4B and 5A.

– It is confusing how the p-values are indicated in Figure Legends for all the panels at once. Would the authors consider to add the p-values to each individual panel?

– Figure 2B. Would the authors explain how the qPCR are normalized?

– Fig2E. FISH image: What is the isoform detected by the probes?

The Scale bar in Hepa1-6 panel is missing, please, add this info accordingly. Also, Ppib housekeeping gene should be mentioned in the figure legend.

– Gels/Western Blots: the authors should consider adding molecular-weight size marker to PCR gels and Western Blots panels in Figure 3A, 4C, 5D, 5E, 5F, 7B and 7E. Also, adding numbers to the lanes will help the reader

– Page 9. Would the authors consider to change "…where the expression was correlated with time points of concerted cell division, but did not fluctuate across the cell cycle" to "…where the expression was correlated with time points of concerted DNA synthesis, but did not fluctuate across the cell cycle".

– The authors claim to visualize single molecule RNA by FISH and quantitate foci per cell. Could the authors explain how they confirm single molecule detection resolution?

– Figure 2E and 2G. How the authors reconcile the differences in PHAROH intracellular distribution by using FISH (2E right panel) and cellular fractionation (2G PHAROH panel)?

– Figure 3. To evaluate the functional role of PHAROH the authors generate targeted knockouts using CRISPR/Cas9 technology, however, they get clones where the expression level is reduced up to 80-90%. Can the authors speculate on why they were not able to get a complete KO? Would the authors consider changing the word "knockout" to "knockdown" in the text, accordingly?

– The authors express PHAROH ectopically in order to rescue cell proliferation and migration defects in PHAROH knockdown cells. How are the levels of this expression controlled? Is this an overexpression of PHAROH in comparison to endogenous levels? Could the authors elaborate more in the interpretation of this data and in the possibility of having other secondary effects due to overexpression. What is the phenotype if the mutant m4 PHAROH is used instead of the wildtype form of PHAROH?

– Figure 4C. Would the authors consider to include a quantification of the western blot lanes?

– Figure 5B and 5E. It would be nice if the authors could indicate in Figure 5B what are the TIAR binding sites used in the EMSA (Figure 5E).

– Figure 7D and 7E. The authors overexpress PHAROH and m4 PHAROH to study translation using c-MYC 3'UTR reporter RNA. Could the authors comment what is the interpretation of the results given the presence of endogenous levels of PHAROH in the cells? It would be more appropriated if the authors could use PHAROH knockdown cells (or antisense) for this study. They should consider studying the effect of PHAROH on c-MYC translation without the contribution of endogenous levels that would compete for TIAR binding.

– Due to the clinical relevance of this study, would the authors consider visualizing LINC00862 lncRNA in resection samples from HCC patients by FISH?

*Reviewer #3:*

In this manuscript, the authors identified a novel lncRNA, named PHAROH, which is expressed in ESC as well as HCCs. Depletion of PHAROH either by genome editing or ASO reduced cell proliferation, which was rescued by exogenous expression of PHAROH. Genes that exhibited altered expression in PHAROH depleted cells harbored consensus upstream sequences, which matches the MYC-binding motifs. The protein expression of MYC was downregulated in PHAROH depleted cells without changes of the mRNA expression of MYX, suggesting that PHAROH regulates MYC expression at the post-transcriptional levels. The authors performed RAP-MS and identified TIAR as a binding partner of PHAROH, and confirmed the interaction by a series of in vitro binding assays. Considering TIAR has been shown to inhibit translation of target mRNAs, the authors proposed that PHAROH normally sequester TIAR to keep the expression of MYC at a high levels in ESC or HCC.

Overall the experiments are well designed and results are clear and convincing. I only have a few minor comments regarding the identities of PHAROH and insights into the molecular mechanisms.

1. While the authors described that there are two isoforms of PHAROH, the statement is not well supported by experimental data. Considering that this is the first paper that characterized the function of PHAROH, more detailed descriptions how they determined the two transcriptional units are required. Northern blot analyses will be the best, but it will be also be helpful to show the mapping patterns of deep sequencing reads together with RACE clones they identified. The annotations shown in Figure S2F is somewhat confusing, which is inconsistent with the annotation shown in Figure 2A. In addition, there are no explanation for the label A, B, C… in Figure S2F.

2. Given that TIAR is a general translational regulators, it is interesting to know how PHAROH specifically regulate the expression of MYC (may not be specific, but the enrichment of MYC binding sequences in the upstream regions of affected genes are extraordinary). I think the clarification of this mechanism may be beyond the scope of this study, but short discussion by these authors will be helpful.

3. Similarly, there is no discussion from the viewpoint of stoichiometry. Because the authors performed PHAROH FISH (Fig2E) and immunostaining of TIAR (FigS7F), I am curious to know if these signals overlap either in the cytoplasm of nucleus. Additional discussion based on the subcellular localization of PHAROH and TIAR will be required.

---

## [Author Response]

Reviewer #1:In this work, the authors attempt to determine how LNC RNAs may be involved in oncogenic progression, in this particular case, hepatocellular carcinoma (HCC).They find, using several databases, that several LNC RNAs are dysregulated and choose the highest one, in 13% of the cases to investigate in detail for mechanistic correlation with the pathology. In a straightforward analysis, they determine that this particular LNC, which they term PHAROH plays a role in cell proliferation, as its knockout and knockdown leads to a decrease in cell proliferation in culture, which is restored by transfection into the cells. Comparing transcriptomics of control and knockout cells, a large number of mRNAs are affected, but they determine that c-Myc promoter elements are enriched in the genes identified, although c-Myc is not among them. RNA pull-down followed by mass spec identifies a protein, TIAR, a known translational repressor that binds to several short sequences in the LNC RNA. The authors postulate that the role of PHAROH is to efficiently compete for binding of the translational repressor with Myc so that c-Myc translation is enhanced without affecting the mRNA levels. Mutation of these binding sites results in decreased proliferation, and importantly oncogenesis. The data supports this model as far as it goes in the cultured cells but whether it plays a role in HCC is yet to be determined. The correlations with cirrhotic human liver seems more compelling in Figure 1D. The focus on PHAROH and TIAR, while revealing may represent only a small part of the total picture and c-Myc one of many possible targets that influence cell proliferation. Nonetheless, this work provides convincing evidence for the model where a LNC RNA can serve as a competitor for a translation regulator and a starting point for disease relevance.The work is convincing within the context of the experimental design involving these particular players selected from a large number of possibilities. While it would not have been possible to investigate many of the other LNCs, proteins, targets, promoter elements etc., the manuscript needs to have the usual caveats in the text concerning the limitations on interpreting its relevance for a complex process such as cancer progression into HCC. The authors have not overextended their claims but they do suggest that PHAROH might be a therapeutic target. Hopefully they're right, but many of these approaches have not played out as expected, due to the large heterogeneity of tumor cells, herein supported by the fact that PHAROH is not overexpressed in the majority of the samples. Whether it is a small pathway in a random subset of cells, or an important one remains to be determined.

As this reviewer points out, we have not overextended our claims, and we have provided caveats where appropriate within the manuscript.

Reviewer #2:– Statistical analysis and P-values. The authors should add p-values to Figures: 2C, 2G, 4B and 5A.– It is confusing how the p-values are indicated in Figure Legends for all the panels at once. Would the authors consider to add the p-values to each individual panel?

We have added the p-values to Figures2C and 4B. In panel 2G, we are only presenting the distribution of the RNA after fractionation, and in panel 5A, we are not comparing the differences between the pulldown efficiency of each oligo. Hence, no p-values are provided.

– Figure 2B. Would the authors explain how the qPCR are normalized?

The qRT-PCR is normalized using *PPIB* mRNA levels, which did not vary greatly among the samples. Normalization is outlined in the methods.

– Fig2E. FISH image: What is the isoform detected by the probes?The Scale bar in Hepa1-6 panel is missing, please, add this info accordingly. Also, Ppib housekeeping gene should be mentioned in the figure legend.

The majority of probes are complementary to the last exon of PHAROH, and thus detects both *PHAROH* isoforms. The scale bar in the Hepa1-6 panel has been added, and the Ppib housekeeping gene is added to the Figure legend.

– Gels/Western Blots: the authors should consider adding molecular-weight size marker to PCR gels and Western Blots panels in Figure 3A, 4C, 5D, 5E, 5F, 7B and 7E. Also, adding numbers to the lanes will help the reader

Molecular-weight size markers to PCR gels were added. We did not provide molecular-weight size markers in the western blot panels as we only blot for one protein using a well characterized antibody.

– Page 9. Would the authors consider to change "…where the expression was correlated with time points of concerted cell division, but did not fluctuate across the cell cycle" to "…where the expression was correlated with time points of concerted DNA synthesis, but did not fluctuate across the cell cycle".

We have changed this in the text.

– The authors claim to visualize single molecule RNA by FISH and quantitate foci per cell. Could the authors explain how they confirm single molecule detection resolution?

We used the ViewRNA ISH kit which is a method based off of Battich et al.’s work (Nat Methods 10, 1127–1133 (2013). https://doi.org/10.1038/nmeth.2657) and has been validated in numerous studies. In addition, our FPKM values from RNA-seq as well as qRT-PCR estimates correlate well with our imaging data.

– Figure 2E and 2G. How the authors reconcile the differences in PHAROH intracellular distribution by using FISH (2E right panel) and cellular fractionation (2G PHAROH panel)?

By FISH and cellular fractionation, PHAROH is 50% cytoplasmic and 50% nuclear (nucleoplasm + chromatin bound). We realize that the normalization may have been confusing to readers and we have updated it accordingly.

– Figure 3. To evaluate the functional role of PHAROH the authors generate targeted knockouts using CRISPR/Cas9 technology, however, they get clones where the expression level is reduced up to 80-90%. Can the authors speculate on why they were not able to get a complete KO? Would the authors consider changing the word "knockout" to "knockdown" in the text, accordingly?

The strategy of our CRISPR/Cas9 knockout involves the deletion of 788 nucleotides around the transcription start site of the gene. Our qRT-PCR primers are located at the 3’ end of the transcript, and thus it is possible that an aberrant transcription state site is activated in some cells. This seems to be a rather general feature of many different lncRNA knockouts.

– The authors express PHAROH ectopically in order to rescue cell proliferation and migration defects in PHAROH knockdown cells. How are the levels of this expression controlled? Is this an overexpression of PHAROH in comparison to endogenous levels? Could the authors elaborate more in the interpretation of this data and in the possibility of having other secondary effects due to overexpression. What is the phenotype if the mutant m4 PHAROH is used instead of the wildtype form of PHAROH?

The rescue is via transient transfection and as such the rescue expression level is higher than the endogenous level of the RNA. We used the m4 mutant in a luciferase assay and Myc blot (Figure 7D, E), however, we did not use it in the rescue assay.

– Figure 4C. Would the authors consider to include a quantification of the western blot lanes?

We’ve added the quantification of the western blot as Figure 4-supplement 4E.

– Figure 5B and 5E. It would be nice if the authors could indicate in Figure 5B what are the TIAR binding sites used in the EMSA (Figure 5E).

The TIAR binding sites are boxed in red in Figure 5B. We assume that *PHAROH* is completely occupied by TIAR in Figure 5E (left). The *PHAROH* mutants (m1-m4) were cumulatively made in a 5’ to 3’ manner. This has also been clarified in the figure legend. Sequences for the mutants are available in the key resources table.

– Figure 7D and 7E. The authors overexpress PHAROH and m4 PHAROH to study translation using c-MYC 3'UTR reporter RNA. Could the authors comment what is the interpretation of the results given the presence of endogenous levels of PHAROH in the cells? It would be more appropriated if the authors could use PHAROH knockdown cells (or antisense) for this study. They should consider studying the effect of PHAROH on c-MYC translation without the contribution of endogenous levels that would compete for TIAR binding.

As compared to the transiently expressed level of *PHAROH*, the endogenous level is low and as such, we expect minimal impact on the assay.

– Due to the clinical relevance of this study, would the authors consider visualizing LINC00862 lncRNA in resection samples from HCC patients by FISH?

This is an interesting point but is currently beyond the scope of this study. However, we have assayed a number of human HCC cell lines for LINC00862 and have included it in Figure 1E.

Reviewer #3:[…] Overall the experiments are well designed and results are clear and convincing. I only have a few minor comments regarding the identities of PHAROH and insights into the molecular mechanisms.1. While the authors described that there are two isoforms of PHAROH, the statement is not well supported by experimental data. Considering that this is the first paper that characterized the function of PHAROH, more detailed descriptions how they determined the two transcriptional units are required. Northern blot analyses will be the best, but it will be also be helpful to show the mapping patterns of deep sequencing reads together with RACE clones they identified. The annotations shown in Figure S2F is somewhat confusing, which is inconsistent with the annotation shown in Figure 2A. In addition, there are no explanation for the label A, B, C… in Figure S2F.

RNA-seq mapping patterns have been overlaid onto Figure 2A. Explanation for label A, B, C has been added in the Figure legends.

2. Given that TIAR is a general translational regulators, it is interesting to know how PHAROH specifically regulate the expression of MYC (may not be specific, but the enrichment of MYC binding sequences in the upstream regions of affected genes are extraordinary). I think the clarification of this mechanism may be beyond the scope of this study, but short discussion by these authors will be helpful.

There was a short discussion of how PHAROH can regulate the expression of MYC in the second to last paragraph of the discussion.

3. Similarly, there is no discussion from the viewpoint of stoichiometry. Because the authors performed PHAROH FISH (Fig2E) and immunostaining of TIAR (FigS7F), I am curious to know if these signals overlap either in the cytoplasm of nucleus. Additional discussion based on the subcellular localization of PHAROH and TIAR will be required.

There is a brief discussion in the Results section regarding the relative abundances of PHAROH and c-MYC mRNA and the feasibility of a competition model.